



# Uncertainty of continuous $\Delta$CO-based $\Delta$ffCO$_2$ estimates derived from $^{14}$C flask and bottom-up $\Delta$CO/$\Delta$ffCO$_2$ ratios

Fabian Maier[1], Ingeborg Levin[1], Sébastien Conil[2], Maksym Gachkivskyi[1,3], Hugo Denier van der Gon[4],
and Samuel Hammer[1,3]

[1]Institut für Umweltphysik, Heidelberg University, INF 229, 69120 Heidelberg, Germany
[2]ANDRA, DISTEC/EES, Observatoire Pérenne de l'Environnement, 55290 Bure, France
[3]ICOS Central Radiocarbon Laboratory, Heidelberg University, Berliner Straße 53, 69120 Heidelberg, Germany
[4]Department of Climate, Air and Sustainability, TNO, Princetonlaan 6, 3584 CB Utrecht, the Netherlands

*Correspondence to*: Fabian Maier (Fabian.Maier@iup.uni-heidelberg.de)

**Abstract.** Measuring the $^{14}$C/C depletion in atmospheric CO$_2$ compared to a clean-air reference is the most direct way to estimate the recently added CO$_2$ contribution from fossil fuel (ff) combustion ($\Delta$ffCO$_2$) in ambient air. However, since $^{14}$CO$_2$ measurements cannot be conducted continuously nor remotely, there are only very sparse $^{14}$C-based $\Delta$ffCO$_2$ estimates available. Continuously measured tracers like carbon monoxide (CO), which are co-emitted with ffCO$_2$ can be used as
additional alternative proxies for $\Delta$ffCO$_2$, provided that the $\Delta$CO/$\Delta$ffCO$_2$ ratios can be determined correctly. Here, we use almost 350 $^{14}$CO$_2$ measurements from flask samples collected between 2019 and 2020 at the urban site Heidelberg in Germany, and corresponding analyses from more than 50 afternoon flasks collected between September 2020 and March 2021 at the rural ICOS site Observatoire pérenne de l'environnement (OPE) in France, to calculate average $\Delta$CO/$\Delta$ffCO$_2$ ratios for those sites. By dividing the hourly $\Delta$CO excess observations by the averaged flask ratio, we construct continuous and bias-free $\Delta$CO-
based $\Delta$ffCO$_2$ records. The comparison between $\Delta$CO-based $\Delta$ffCO$_2$ and $^{14}$C-based $\Delta$ffCO$_2$ from the flasks yields a root-mean-square deviation (RMSD) of about 4 ppm for the urban site Heidelberg and of 1.5 ppm for the rural site OPE. While for OPE this uncertainty can be explained by observational uncertainties alone, for Heidelberg about half of the uncertainty is caused by the neglected variability of the $\Delta$CO/$\Delta$ffCO$_2$ ratios. We further show that modelled ratios based on a bottom-up European emission inventory would lead to substantial biases in the $\Delta$CO-based $\Delta$ffCO$_2$ estimates for Heidelberg, and also for OPE.
This highlights the need for an ongoing observational calibration/validation of inventory-based ratios, if those shall be applied for large-scale $\Delta$CO-based $\Delta$ffCO$_2$ estimates, e.g. from satellites.



## 1 Introduction

The observational separation of the fossil fuel $CO_2$ contributions ($\Delta ffCO_2$) in regional atmospheric $CO_2$ excess is a prerequisite for independent "top-down" evaluation of bottom-up $ffCO_2$ emission inventories (Ciais et al., 2016). The most direct method for estimating regional $\Delta ffCO_2$ contributions is measuring the ambient air $\Delta^{14}CO_2$ depletion compared to a clean air $\Delta^{14}CO_2$ reference, as fossil fuels are devoid of $^{14}C$, which has a half-life of 5700 years (Currie, 2004; for the $\Delta^{14}CO_2$ notation see Stuiver and Polach, 1977). Many studies have successfully applied this approach to directly estimate regional $\Delta ffCO_2$

concentrations in urban regions (Levin et al., 2003; Levin and Rödenbeck, 2008; Turnbull et al., 2015; Zhou et al., 2020), which could then be used in atmospheric inverse modeling systems to compare with bottom-up $ffCO_2$ emission inventories (Graven et al., 2018; Wang et al., 2018). One drawback of this method is, however, that $^{14}C$-based $\Delta ffCO_2$ estimates have typically only a low (i.e. weekly or monthly) temporal resolution and poor spatial coverage, due to the labor-intensive and costly process of collecting and measuring precisely air samples for $^{14}CO_2$. Up to now, $^{14}CO_2$ observations cannot be conducted

continuously with the precision needed for atmospheric $\Delta ffCO_2$ determination neither can $^{14}CO_2$ observations be performed remotely, e.g. with satellites. This limits the potential of $^{14}C$ observations to estimate $ffCO_2$ emissions at the continental scale and at high spatiotemporal resolution.

Therefore, more frequently measured gases like carbon monoxide (CO), which is co-emitted with $ffCO_2$ during incomplete

combustion, are being used as additional constraint for estimating $ffCO_2$ emissions (e.g., Palmer et al., 2006; Boschetti et al., 2018). Also, CO observations from satellites showed high potential for verifying and optimizing bottom-up $ffCO_2$ emission estimates of large industrial regions in the whole world (Konovalov et al., 2016). However, using CO observations in inverse models for estimating $ffCO_2$ emissions requires decent information about the spatiotemporal variability of the $CO/ffCO_2$ emission ratios. Typically, this information is taken from bottom-up CO and $CO_2$ emission inventories, which are based on

national activity data and source sector specific emission factors (Janssens-Maenhout et al., 2019; Kuenen et al., 2022). However, these emission factors are associated with high uncertainties, especially for CO, since they strongly depend on the often variable combustion conditions (Dellaert et al., 2019). Observation-based verification of the bottom-up emission ratios may significantly reduce biases in top-down $ffCO_2$ emission estimates.

Continuously measured $\Delta CO$ offsets compared to a clean air reference were used in the past to construct temporally highly resolved $\Delta ffCO_2$ concentration records, which can provide additional spatiotemporal information for constraining fossil emissions in transport model inversions. For this, the continuous $\Delta CO$ measurements are divided by mean $\langle \Delta CO/\Delta ffCO_2 \rangle$ ratios, which are representative for the particular observation site and the averaging period (Gamnitzer et al., 2006; Levin and Karstens, 2007; van der Laan et al., 2010; Vogel et al., 2010). Note that we use in this study the "$\Delta$" in front of "CO" and

"$ffCO_2$" to describe the excess concentrations compared to the clean air reference; it is different from the $\Delta$-notation introduced by Stuiver and Polach (1977) to report the $^{14}CO_2$ measurements. At observation sites with simultaneous $^{14}CO_2$ measurements



the $<\Delta CO/\Delta ffCO_2>$ ratios can be calculated from $^{14}C$-based $\Delta ffCO_2$ estimates. This allows to calculate continuous $\Delta CO$-based $\Delta ffCO_2$ concentration offsets, which are fully independent of bottom-up emission information. For example, Vogel et al. (2010) used weekly integrated $\Delta^{14}CO_2$ observations combined with occasional hourly $\Delta^{14}CO_2$ flask data from the urban site

Heidelberg, located in a heavily industrialized area in the Upper Rhine Valley in Southwestern Germany, to estimate continuous $\Delta CO$-based $\Delta ffCO_2$ concentrations. They show that calculating the $\Delta CO/\Delta ffCO_2$ ratios from the weekly integrated $\Delta^{14}CO_2$ samples leads to biases in the $\Delta CO$-based $\Delta ffCO_2$ estimates, since the weekly averaged ratios are biased towards hours with high $\Delta ffCO_2$ concentrations. That is why they used the $\Delta^{14}CO_2$ flask data to calculate mean diurnal cycles for the summer and winter period. By correcting the weekly averaged ratios with these diurnal profiles, they could reduce some of the bias of

the $\Delta CO$-based $\Delta ffCO_2$ estimates.

## 1.1 Research question and objectives

To bring this approach further, we have collected almost 350 hourly-integrated $\Delta^{14}CO_2$ flask samples during 2019 and 2020 with very different atmospheric conditions at the Heidelberg observation site. The purpose of this high-frequent flask sampling is to investigate if such a large flask pool allows an estimation of the urban $ffCO_2$ emissions in the Heidelberg footprint (see

the companion paper by Maier et al., 2023a). Our aim in the present study is to assess the use of these hourly $\Delta^{14}CO_2$ flask data to estimate $\Delta CO/\Delta ffCO_2$ ratios and then derive a continuous $\Delta CO$-based $\Delta ffCO_2$ record for the Heidelberg station. We further estimate the uncertainty of this $\Delta CO$-based $\Delta ffCO_2$ record and assess the share of uncertainty that is caused by the spatiotemporal variability of the emission ratios in Heidelberg's surrounding sources. To test this approach at a more remote site, a similar investigation is conducted at a rural Integrated Carbon Observation System (ICOS, Heiskanen et al., 2022)

atmosphere station, Observatoire pérenne de l'environnement (OPE), but using only about 50 hourly integrated flask samples collected between September 2020 and March 2021.

We further compare the flask-based $\Delta CO/\Delta ffCO_2$ ratios with modelled ratios based on bottom-up estimates from the high-resolution emission inventory of the Netherlands Organization for Applied Scientific Research (TNO, Dellaert et al., 2019;

Denier van der Gon et al., 2019). As mentioned above, the observation-based validation of the bottom-up emission ratios is a critical improvement when using CO concentration measurements as an additional tracer in inverse models to estimate $ffCO_2$ emissions. Moreover, this comparison allows to investigate if the modelled, inventory-based $\Delta CO/\Delta ffCO_2$ ratios could be used to construct a $\Delta CO$-based $\Delta ffCO_2$ record at sites without $^{14}CO_2$ measurement. For example, ambient air CO concentrations are frequently measured at urban emission hot spots, as CO emissions affect air pollution and human health (Pinty et al., 2019).

At such sites, using inventory-based $\Delta CO/\Delta ffCO_2$ ratios is thus the only option to calculate continuous $\Delta CO$-based $\Delta ffCO_2$ records, which could play an important role in quantifying anthropogenic $ffCO_2$ emissions in urban hot spot regions. However, this inventory-based approach strongly relies on correct bottom-up CO and $ffCO_2$ emissions. Furthermore, it ignores non-fossil CO sources like biomass burning or CO production due to oxidation of methane and volatile organic compounds (VOCs), and neglects the CO sinks such as the atmospheric oxidation by hydroxyl (OH) radicals (Folberth et al., 2006) and soil uptake





(Inman et al., 1971). Therefore, such an inventory-based approach assumes a neglectable influence from non-fossil CO sources and sinks without proper validation of that assumption.

## 2 Methods

### 2.1 Site and data description

We calculate representative $\Delta CO/\Delta ffCO_2$ ratios for the urban site Heidelberg (49.42°N, 8.67°E) in Southwest Germany and

the rural site OPE (48.56°N, 5.50°E) in Eastern France. Heidelberg is a medium-sized city (~160'000 inhabitants) located in the densely populated Upper Rhine-Valley. As is typical for an urban site, Heidelberg is surrounded by many different anthropogenic $CO_2$ and CO sources, which leads to a large spatial variability of the $CO/ffCO_2$ emission ratios in the footprint of the station. The observation site (30 m a.g.l.) is located on the university campus, and thus local emissions are mainly from the traffic and heating sectors. Furthermore, there is also a combined heat and power plant located to the North at 500 m

distance of the site, as well as two heavily industrialised cities Mannheim and Ludwigshafen, including a large coal-fired power plant and the BASF factory at a 15-20 km distance to the North-West. The OPE station is located on a 400 m altitude hill, mainly surrounded by cropland (Storm et al., 2022), in a much less populated remote rural region, about 250 km East of Paris. The OPE site is a class-1 station in the ICOS atmosphere network and the flask samples are collected from the highest level of a 120 m tall tower.


At both stations, CO is continuously measured with a Cavity Ring-Down Spectroscopy (CRDS) gas analyser (for OPE data see Conil et al., 2019). Furthermore, hourly flask samples are collected at both stations with an automated ICOS flask sampler (see Levin et al., 2020). Thereby, the air flow into the flasks is regulated by mass flow controllers, so that the final air sample in the flasks approximates 1-hour average concentrations of ambient air. In Heidelberg, we sampled very different atmospheric

situations, i.e. during well-mixed conditions in the afternoon, but also during the morning and evening rush-hours and at night, with almost 350 flasks during the two years 2019 and 2020. At OPE, the flask sampler was programmed to fill one flask every third noon between September 2020 and March 2021, so that there are $^{14}CO_2$ results from more than 50 flasks available in this time period. The $CO_2$ and CO concentrations of the collected flask samples are measured at the ICOS Flask and Calibration Laboratory (FCL, https://www.icos-cal.eu/fcl) with a gas chromatographic analysis system (GC). Afterwards, the $CO_2$ in the

flask samples is extracted and graphitized in the Central Radiocarbon Laboratory (CRL, https://www.icos-cal.eu/crl; Lux, 2018), and the $^{14}C$ is analysed with an accelerator mass spectrometer (AMS, Kromer et al., 2013). The $\Delta^{14}CO_2$ measurements are reported in the so-called $\Delta$-notation introduced by Stuiver and Polach (1977), which normalizes for fractionation processes and expresses the $^{14}C/C$ deviation of the sample from a standard activity in ‰. The typical $\Delta^{14}CO_2$ and CO measurement uncertainties for the hourly flasks are better than 2.5‰ and 2ppb, respectively.



### 2.2 Construction of a continuous ΔCO-based ΔffCO₂ record

We construct a continuous $\Delta CO$-based $\Delta ffCO_2$ record relative to marine background air with hourly resolution ($\Delta ffCO_2^{hrly}$) by dividing the hourly $\Delta CO$ concentrations ($\Delta CO^{hrly}$) by an average $^{14}C$ flask-based $\Delta CO/\Delta ffCO_2$ ratio $\langle R_{flask} \rangle$:

$$\Delta ffCO_2^{hrly} = \frac{\Delta CO^{hrly}}{\langle R_{flask} \rangle} \tag{1}$$

To calculate the $\Delta CO$ and the $^{14}C$-based $\Delta ffCO_2$ excess concentrations at the Heidelberg and OPE observation sites, we must choose an appropriate CO and $\Delta^{14}CO_2$ background. Back-trajectory analyses by Maier et al. (2023b) show a predominant westerly influence for stations in Central Europe; about 2/3 of all back-trajectories, which were calculated for nine European ICOS sites for the full year 2018, end over the Atlantic Ocean at the western boundary of the European continent. Indeed, we identified Mace Head (MHD, 53.33°N, 9.90°W, 5 m a.s.l.), which is located at the west coast of Ireland, to be an appropriate marine reference site for Central Europe. Therefore, we use smooth fit curves through weekly CO flask results (Petron et al., 2022) and two-week integrated $\Delta^{14}CO_2$ samples from MHD as our CO and $\Delta^{14}CO_2$ background, respectively. The applied curve fitting algorithm was developed by the National Oceanic and Atmospheric Administration (NOAA, Thoning et al., 1989). This algorithm yields a fit standard deviation of 13.37 ppb and 0.86‰, respectively, for the CO and $\Delta^{14}CO_2$ background curves.

Obviously, MHD is a less representative background station for situations with non-western air masses. For this, Maier et al. (2023b) estimated a representativeness bias and uncertainty for the MHD background of 0.09±0.28 ppm $ffCO_2$ for hourly flask samples collected in Central Europe at the eastern ICOS site Křešín. We assume this estimate to be an upper limit for the Heidelberg and OPE sites, which are located further to the West. Therefore, we decided to neglect the representativeness bias in our calculations. However, we take into account its variability, which is the representativeness uncertainty of the MHD background. The 0.28 ppm $ffCO_2$ uncertainty would result in a representativeness uncertainty of 0.64‰ for the MHD $\Delta^{14}CO_2$ background, if one assumes that a 1 ppm $ffCO_2$ signal is caused by a 2.3‰ $\Delta^{14}CO_2$ depletion (we deduced this conversion factor from the Heidelberg flask results). Similarly, we can estimate the representativeness uncertainty for the CO background, if we assume a mean $CO/ffCO_2$ ratio to convert the estimated 0.28 ppm $ffCO_2$ uncertainty into a CO uncertainty. The TNO inventory suggests for the Eastern boundary of our model domain (within 22-23°E and 37-61°N) a mean $CO/ffCO_2$ emission ratio of roughly 18 ppb/ppm in 2020. We use this ratio as an upper limit and get a CO background representativeness uncertainty of 0.28 ppm*18 ppb/ppm=5.04 ppb. To estimate the overall CO and $\Delta^{14}CO_2$ background uncertainty, we add the fit uncertainty and the representativeness uncertainty quadratically, which yields 14.29 ppb and 1.07‰, respectively.

### 2.2.1 Calculation of an observation-based $\langle R_{flask} \rangle$ ratio

To calculate the $^{14}C$-based $\langle R_{flask} \rangle$ ratio, we first estimate the $\Delta ffCO_2$ concentrations from the $\Delta^{14}CO_2$ difference between the Heidelberg and OPE measurements and smoothed fits through the MHD background data. For this, we use the following Eq.



2 from Maier et al. (2023b), which also contains a correction for contaminating $^{14}CO_2$ emissions from nuclear facilities and for the potentially $^{14}$C-enriched $\Delta^{14}CO_2$ signature of biosphere respiration (still releasing stored nuclear bomb $^{14}CO_2$ to the atmosphere):

$$\Delta ffCO_2 = C_{bg} \cdot \frac{\Delta^{14}_{bg} - \Delta^{14}_{meas}}{\Delta^{14}_{meas} + 1000\text{‰}} + C_{meas} \cdot \frac{\Delta^{14}_{nuc}}{\Delta^{14}_{meas} + 1000\text{‰}} + C_{resp} \cdot \frac{\Delta^{14}_{resp} - \Delta^{14}_{meas}}{\Delta^{14}_{meas} + 1000\text{‰}} \tag{2}$$

Table 1 shows a compilation of all components of Eq. 2 with short explanations. In general, we used the same procedure as described by Maier et al. (2023b) to estimate the correction terms in Eq. 2. Note, that we only use the flask results with a modelled nuclear contamination below 2‰, to avoid huge nuclear corrections whose uncertainty exceeds the typical uncertainty of the $\Delta^{14}CO_2$ measurements (see Maier et al., 2023b).

**Table 1: Description of the components in Eq. 2.**

| Component | Description | Method |
|---|---|---|
| $\Delta^{14}_{meas}$ | $\Delta^{14}CO_2$ of flasks from observation site | Measured |
| $\Delta^{14}_{bg}$ | $\Delta^{14}CO_2$ background curve | NOAA fit through integrated samples from MHD |
| $\Delta^{14}_{nuc}$ | $\Delta^{14}CO_2$ contamination from nuclear facilities | Modelled using WRF-STILT in combination with nuclear $^{14}CO_2$ emissions from the Radioactive Discharges Database RADD (see Maier et al., 2023b) |
| $\Delta^{14}_{resp}$ | $\Delta^{14}CO_2$ signature of biosphere respiration | Modelled based on Naegler and Levin (2009) |
| $C_{meas}$ | $CO_2$ concentration of flasks from observation site | Measured |
| $C_{bg}$ | $CO_2$ background curve | NOAA fit through weekly flasks from MHD (Lan et al., 2022) |
| $C_{resp}$ | $CO_2$ signal from respiration | Modelled with the Vegetation Photosynthesis and Respiration Model (VPRM, Mahadevan et al., 2008) coupled to STILT |

We then use the weighted total least squares algorithm from Wurm (2022) to calculate regression lines to the $\Delta$CO and $^{14}$C-based $\Delta ffCO_2$ concentrations of the 343 flasks from Heidelberg and the 52 flasks from OPE. This regression algorithm is built on the code from Krystek and Anton (2007) and considers uncertainties in both the $\Delta$CO and $\Delta ffCO_2$ flask concentrations. In

this study, we force the regression line through the origin. Thus, we assume that the very well-mixed and clean air masses at





the observation sites without $\Delta ffCO_2$ excess also represent the background CO concentrations at MHD. The slope of the regression line gives then an unbiased estimate of the $<R_{flask}>$ ratio for the corresponding observation site and time period of the flask samples. In appendix A1, we show why one should use a regression algorithm, which considers the uncertainties in the dependent and independent variables, to calculate a mean bias-free $<\Delta CO/\Delta ffCO_2>$ ratio instead of error-weighted means

or median estimates from individual samples.

### 2.2.2 Modelling of inventory-based $\Delta CO/\Delta ffCO_2$ ratios

To compare the $^{14}C$-based $\Delta CO/\Delta ffCO_2$ ratios with inventory-based ratios, we need to weigh the bottom-up emissions with the footprints of the observation sites. For this, we use the following modeling setup to simulate hourly $\Delta CO$ and $\Delta ffCO_2$ excess at the Heidelberg and OPE observation sites. The Stochastic Time-Inverted Lagrangian Transport (STILT) model (Lin

et al., 2003) is coupled with the Weather Research and Forecast (WRF) model (Nehrkorn et al., 2010), which is driven by the high-resolution CO and $ffCO_2$ emission fluxes from the TNO inventories (Dellaert et al., 2019; Denier van der Gon et al., 2019). The WRF-STILT domain expands from 37°N to 61°N and from 10°W to 23°E (see Fig. 1). The input meteorological fields are taken from the European ReAnalysis 5 data from the European Centre for Medium-Range Weather Forecasts (ECMWF) with a horizontal resolution of 0.25° (https://www.ecmwf.int/en/forecasts/datasets/reanalysis-datasets/era5). The

WRF model setup combines an inner domain with a 2 km horizontal resolution along the Rhine Valley (red rectangular in Fig. 1) nested in a larger European domain with a 10 km resolution used to calculate the hourly footprints with STILT. Those footprints are then mapped with the high-resolution CO and $ffCO_2$ emissions from TNO to get the CO and $ffCO_2$ concentrations for Heidelberg and OPE. As we assume zero CO and $ffCO_2$ concentrations at the boundaries of the STILT model domain, the modelled CO and $ffCO_2$ concentrations directly correspond to the $\Delta CO$ and $\Delta ffCO_2$ excesses with respect to the model domain

boundaries. The TNO inventory divides the total CO and $ffCO_2$ emissions into 15 emission sectors with individual monthly, weekly and diurnal temporal emission profiles. Note, that the CO emissions from TNO contain the fossil fuel and biofuel CO contributions. Since Heidelberg is surrounded by many point sources with elevated stacks, we treat the TNO point sources within the Rhine Valley separately with the STILT volume source influence (VSI) approach (see Maier et al., 2022), to model the point source contributions at the Heidelberg site. In this model study, we fully neglect natural CO sources and sinks as well

as atmospheric CO chemistry.



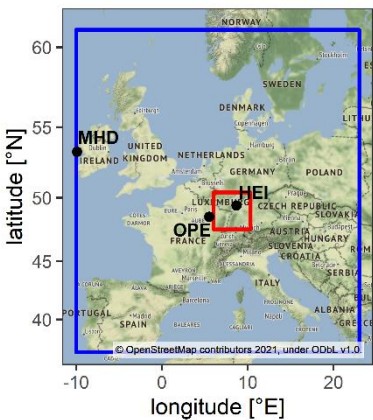

**Figure 1: Map of the European WRF-STILT model domain (framed in blue). The Heidelberg (HEI) and OPE observation sites as well as the Mace Head (MHD) background site are indicated. The red rectangular shows the Rhine Valley domain.**

# 3 Results

## 3.1 Study at the urban site Heidelberg

### 3.1.1 $^{14}$C-based $\Delta CO/\Delta ffCO_2$ ratios from flask samples

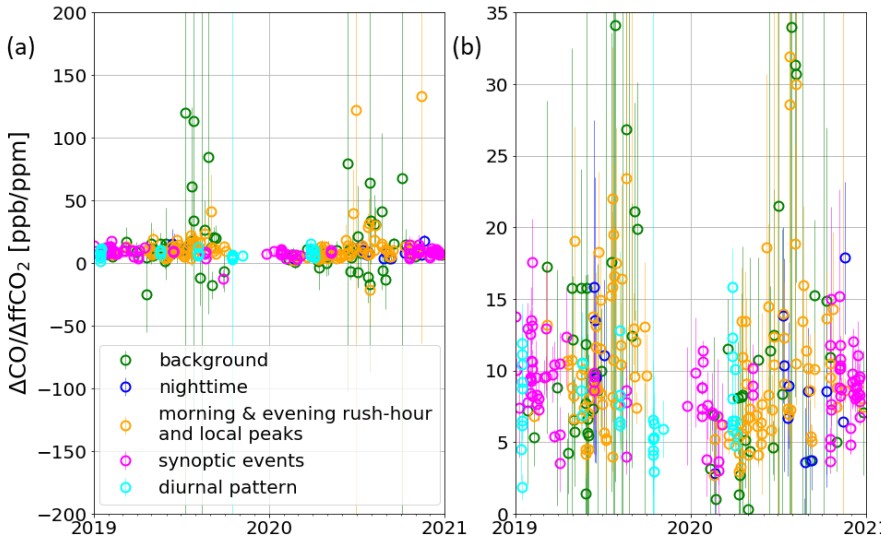

**Figure 2: $^{14}$C-based $\Delta CO/\Delta ffCO_2$ ratios from hourly flasks collected at the Heidelberg observation site between 2019 and 2020. Very different atmospheric conditions are sampled as indicated by the different colors. We sampled almost background conditions (green), $CO_2$ enhancements during night (blue), morning and evening rush-hour peaks and $CO_2$ spikes from most probably local sources (orange), synoptic events with a $CO_2$ concentration built up over several days (magenta) as well as diurnal cycles (cyan). Panel (b) shows a zoom into panel (a).**





Figure 2 shows the $^{14}$C-based $\Delta CO/\Delta ffCO_2$ ratios of the hourly flask samples from Heidelberg, which were collected during very different atmospheric conditions (see colors in Fig. 2). By testing different flask sampling strategies, we sampled very
different situations ranging from almost background conditions, nighttime $CO_2$ enhancements, morning and evening rush-hour signals and local contaminations as well as large-scale synoptic events and diurnal patterns. We observe large positive and also negative ratios with enormous error bars mainly during summer and during well-mixed atmospheric (background) conditions (green dots). These outliers are associated with very low (or even negative) $\Delta ffCO_2$ concentrations and large relative $\Delta ffCO_2$ uncertainties that blow up the ratio and its uncertainty. Indeed, these individual unrealistic ratios can lead to a bias in
the mean or median of the (averaged) ratios, as we show with a synthetic data study in Appendix A1. However, the slope of an error weighted regression through the flask $\Delta CO$ and $\Delta ffCO_2$ excess concentrations represents an un-biased estimate of the $<R_{flask}>$ ratio (see Fig. 3a).

The slope of this regression yields an average ratio of $8.44 \pm 0.07$ ppb/ppm for all flasks collected during the two years 2019
and 2020. The good correlation indicated by an $R^2$ value of 0.88 is predominantly caused by the flasks with large $\Delta CO$ and $\Delta ffCO_2$ concentrations, which were mainly collected during synoptic events in the winter half-year (see Fig. A2). While limiting the analysis to the cold season flasks gives a ratio of $8.52 \pm 0.08$ ppb/ppm with a high correlation ($R^2 = 0.89$), the warm season flasks are associated with a slightly smaller ratio of $8.08 \pm 0.17$ ppb/ppm but much poorer correlation ($R^2 = 0.36$). Thus, there might be a seasonal cycle in the relationships between $\Delta CO$ and $\Delta ffCO_2$, the correlation being strong in the cold
period but much weaker during the warm period. But there is no evidence of a significant seasonal cycle in the ratios, the winter ratio being only 5% larger than the summer ratio. So, the seasonal cycle is not in the ratio by itself but more in its significance or robustness. The daily cycle of the ratios seems as well to be small. The afternoon flasks show an average ratio of $8.60 \pm 0.19$ ppb/ppm ($R^2 = 0.84$) while non-afternoon flasks show an average ratio of $8.41 \pm 0.08$ ppb/ppm ($R^2 = 0.88$). Furthermore, there is only a slightly decreasing trend between 2019 ($8.57 \pm 0.11$ ppb/ppm, $R^2 = 0.87$) and the Covid-19 year
2020 ($8.35 \pm 0.10$ ppb/ppm, $R^2 = 0.88$), which is, however, within the $2\sigma$ uncertainty range.





### 3.1.2 Uncertainty of the ΔCO-based ΔffCO₂ record

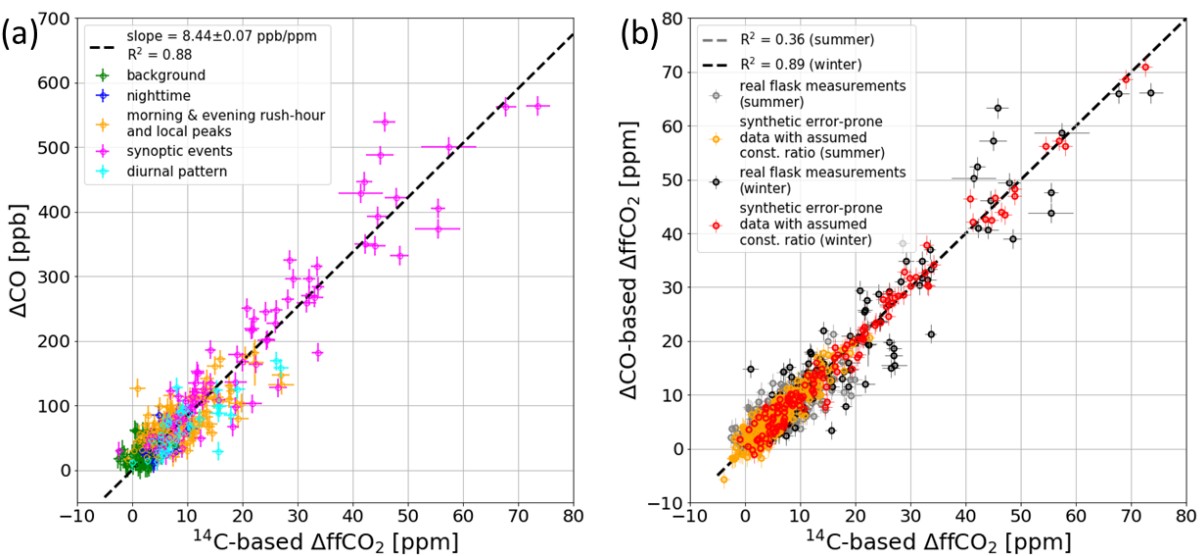

**Figure 3: (a) Scatter plot with the measured ΔCO and the ¹⁴C-based ΔffCO₂ concentrations of the hourly flasks collected at the Heidelberg observation site between 2019 and 2020. The colors indicate the sampling situation of the flasks (see description in the caption of Fig. 2). The black dashed line shows a regression line calculated with the weighted total least squares algorithm from Wurm (2022). (b) Comparison between the ΔCO-based ΔffCO₂ (from Eq. 1) and the ¹⁴C-based ΔffCO₂ concentrations of the Heidelberg winter (black dots) and summer (grey dots) flasks and of the synthetic data (red dots indicate the winter data and orange dots corresponds to the summer data). The synthetic data were generated by assuming a constant ΔCO/ΔffCO₂ ratio, which is the average <ΔCO/ΔffCO₂> ratio from the Heidelberg flasks. Therefore, the scattering of the orange and red data points is only caused by the measurement and background representativeness uncertainties of the ΔCO and ¹⁴C-based ΔffCO₂ concentrations. This means that the increased scattering of the real data (black and grey) compared to the synthetic data (red and orange) is caused by the variability of the ratios.**

Because of the small daily and seasonal differences in the ¹⁴C-based ΔCO/ΔffCO₂ ratios and the difficulty to calculate average summer ratios (see Appendix A1), we decided to use the average ratio of all flasks to compute a continuous hourly ΔCO-based ΔffCO₂ record for the two years 2019 and 2020. However, this means that we fully neglect any spatiotemporal variability in the ratios. At times when we have collected flasks, we can then compare these ΔCO-based ΔffCO₂ estimates with the ¹⁴C-based ΔffCO₂ concentrations of the flasks (see Fig. 3b, black dots). Obviously, a regression through this data yields a slope of 1 since we used the average ratio of all flasks to construct the ΔCO-based ΔffCO₂ record. The (vertical) scattering of the data around this 1:1 line, e.g. the root-mean-square deviation (RMSD) between ΔCO- and ¹⁴C-based ΔffCO₂, can be used as an estimate for the uncertainty of the ΔCO-based ΔffCO₂ record. This RMSD is 3.95 ppm, which is almost 4 times larger than the typical uncertainty for ¹⁴C-based ΔffCO₂. As the RMSD depends on the range of the ΔffCO₂ concentrations, we also compute a normalized RMSD (NRMSD), by dividing the RMSD by the mean ¹⁴C-based ΔffCO₂ concentration of the flasks. This gives a NRMSD of 0.39, which means that the RMSD adds up to 39% to the average ΔffCO₂ excess at Heidelberg.





In the following, we want to assess the sources of this increased uncertainty: Is it mainly caused by the measurement and background representativeness uncertainties of the $\Delta CO$ and $^{14}C$-based $\Delta ffCO_2$ concentrations, or is it rather due to the variability of the ratios that we fully neglect when using a constant ratio to derive the $\Delta CO$-based $\Delta ffCO_2$ record? To answer this, we performed a synthetic data experiment, in which we assumed a "true" constant $\Delta CO/\Delta ffCO_2$ ratio of 8.44 ppb/ppm. We used this constant ratio together with the observed $\Delta ffCO_2$ concentrations from the flasks to create synthetic "true", i.e.,

error-free $\Delta CO$ and $\Delta ffCO_2$ data pairs (see Appendix A1). We then drew random numbers from an unbiased Gaussian distribution with a $1\sigma$ range of 1.16 ppm (for $\Delta ffCO_2$) and 14.49 ppb (for $\Delta CO$), which represents the mean $\Delta CO$ and $\Delta ffCO_2$ uncertainties of the real measurements (see Sect. 2.2). These random numbers were added to the synthetic "true" $\Delta CO$ and $\Delta ffCO_2$ data to get error-prone synthetic $\Delta CO$ and $\Delta ffCO_2$ concentrations. After that, we used the error-prone synthetic $\Delta CO$ data and the $\Delta CO/\Delta ffCO_2$ ratio of 8.44 ppb/ppm to calculate synthetic $\Delta CO$-based $\Delta ffCO_2$ concentrations. By comparing the

error-prone synthetic $\Delta ffCO_2$ concentrations with the synthetic $\Delta CO$-based $\Delta ffCO_2$ concentrations, we get a lower RMSD of only 2.07 ppm. By construction, this synthetic data experiment covers the same $\Delta CO$ and $\Delta ffCO_2$ ranges like the real flask observations but assumes a constant ratio. Therefore, the difference between the RMSD of the real $\Delta ffCO_2$ observations (3.95 ppm) and the synthetic data (2.07 ppm) must be caused by the variability of the ratios. Thus, about half of the uncertainty of the $\Delta CO$-based $\Delta ffCO_2$ record can be attributed to uncertainties of the $\Delta CO$ and $\Delta ffCO_2$ excess concentrations, and the

remaining half of this uncertainty originates from the variability of the ratios.

### 3.1.3 Comparison of observed with inventory-based $\Delta CO/\Delta ffCO_2$ ratios

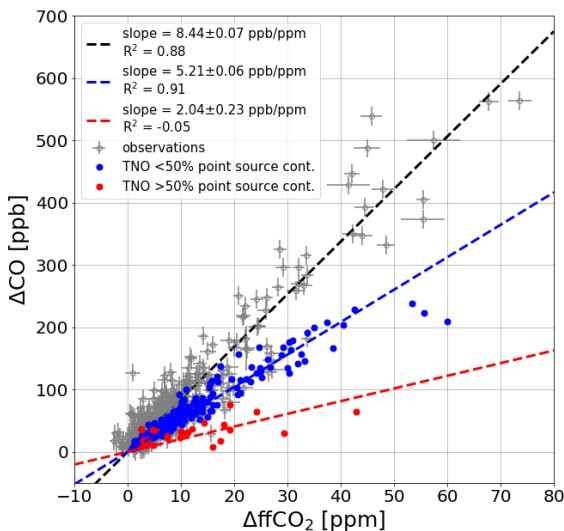

**Figure 4: WRF-STILT simulation of the TNO $\Delta CO$ and $\Delta ffCO_2$ contributions in Heidelberg, 30m, from emissions within the European STILT domain (see Fig. 1). The model results are shown only for hours with flask sampling events between 2019 and**
**2020. The blue and red points indicate hourly situations with a point source contribution of less and more than 50%, respectively. As a reference, the flask observations are also shown in grey. The dashed lines show linear regressions through the respective data points.**



We also compared our flasks [14]C-based $\Delta CO/\Delta ffCO_2$ ratios with the high-resolution emission inventory from TNO. We simulated the hourly $\Delta CO$ and $\Delta ffCO_2$ contributions in Heidelberg, by transporting the CO and $ffCO_2$ emissions from the TNO inventory over the European STILT domain (see Fig. 1). Figure 4 shows, for the flask sampling events in 2019 and 2020, the respective simulated $\Delta CO$ and $\Delta ffCO_2$ results (red and blue dots). In contrast to the flask observations (grey crosses), the simulated data do not scatter around a single regression line corresponding to a constant ratio. The model results rather show two branches indicating two different ratios. If the contributions from point sources in the simulated $\Delta ffCO_2$ is larger than 50% (red points in Fig. 4), the data scatter around a regression line with slope $2.04 \pm 0.23$ ppb/ppm and poor correlation ($R^2$=-0.05). But, if the contributions from point sources in the simulated $\Delta ffCO_2$ is below 50% (blue points in Fig. 4), the data yield a ratio of $5.21 \pm 0.06$ ppb/ppm with a good correlation ($R^2$=0.91).

This comparison with the flask observations highlights two complementary findings. First, the Heidelberg observation site is rarely influenced by events with strong point source contributions larger than 50% because hardly any of the observed ratio scatters around the red regression line in Fig. 4 and thus shows a point source dominated ratio (that lies around 2 ppb/ppm). The model results for Heidelberg thus often overestimate the contributions from point sources. Second, the area source dominated model results with point source contributions smaller than 50% show an equally high correlation as the observations. This indicates that the emission ratios for the dominating heating and traffic sectors are probably currently very similar in the main footprint of Heidelberg. However, the 5.2 ppb/ppm ratio found in the model results is almost 40% lower compared to the 8.44 ppb/ppm observed ratio. Furthermore, the contributions from the area source emissions alone lead to an average ratio of $6.02 \pm 0.06$ ppb/ppm. This might indicate that TNO underestimates the ratios of the area source emissions in the Rhine Valley. This finding is striking because it means that, consequently, inventory-based ratios would lead to large biases if they were used to calculate a $\Delta CO$-based $\Delta ffCO_2$ record for Heidelberg. It underlines the added value of the station-based observations and the necessary support for long term monitoring. In the following, we present the results of our study performed at the rural ICOS site OPE.





## 3.2 Study at the rural ICOS site OPE

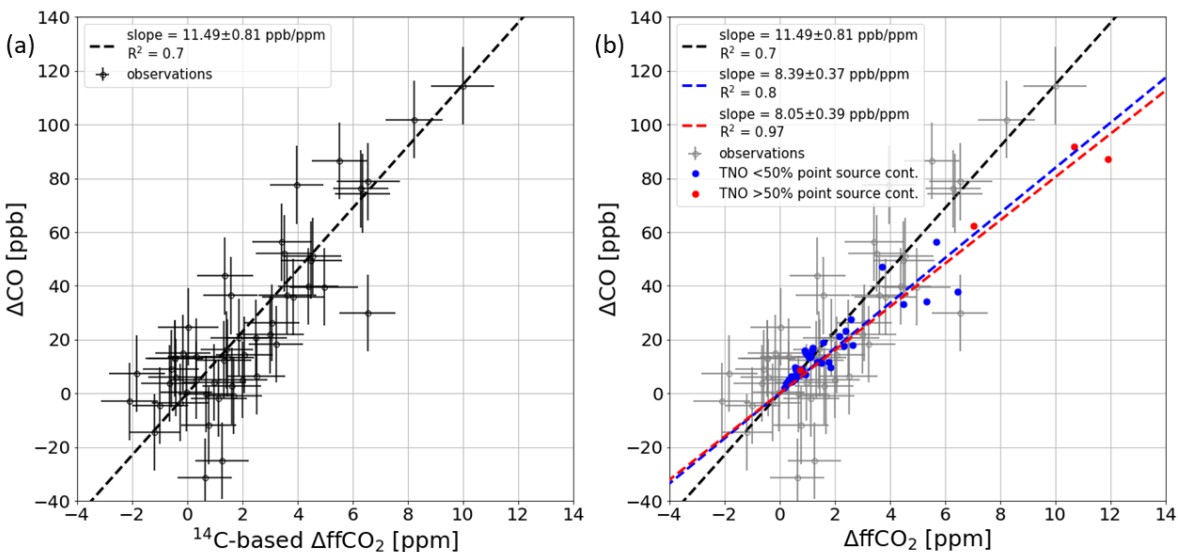

**Figure 5: (a) ΔCO and ΔffCO₂ concentrations from hourly afternoon flasks collected at the OPE station between September 2020 and March 2021. The black dashed line shows a regression line performed with the weighted total least squares algorithm from Wurm (2022). (b) WRF-STILT simulation of the TNO ΔCO and ΔffCO₂ contributions at OPE from emissions within the European STILT domain (see Fig. 1) for the hours with flask sampling events at OPE. The blue and red points indicate hourly situations with a point source contribution of less and more than 50%, respectively. As a reference, the flask observations are also shown in grey. The dashed lines show linear regressions through the respective data points.**

We now want to investigate if the flask observations from a remote site can be used for estimating a continuous ΔCO-based ΔffCO₂ record. Figure 5a shows the ΔCO and $^{14}$C-based ΔffCO₂ observations of 52 flasks from the OPE station, which were collected nearly every third day between September 2020 and March 2021 in the early afternoon. The flasks have an average ΔffCO₂ concentration of 2.19 ppm showing that OPE is much less influenced by polluted air masses compared to the urban site Heidelberg. The regression algorithm from Wurm (2022) gives an average flask ΔCO/ΔffCO₂ ratio of 11.49 ± 0.81 ppb/ppm ($R^2$ = 0.70), which is 3 ppb/ppm larger than the average ratio observed in Heidelberg during 2019 and 2020. Furthermore, the 1σ uncertainty of the slope of the regression line is 10 times larger compared to Heidelberg. This comes along with a reduced correlation between ΔCO and ΔffCO₂ and can at least partly be explained by the smaller range of ΔffCO₂ concentrations sampled at OPE (see Appendix A1). Since all flasks were collected in the winter half-year and during the afternoon, it is not possible to draw conclusions about potential seasonal or diurnal cycles in the ΔCO/ΔffCO₂ ratios at OPE.

Again, we want to use this estimated ratio from the collected flasks to calculate with Eq. 1 an hourly ΔCO-based ΔffCO₂ record for OPE. The RMSD between the ΔCO-based ΔffCO₂ and the $^{14}$C-based ΔffCO₂ from the flasks is only 1.49 ppm, which is due to the much smaller ΔffCO₂ excess at OPE compared to Heidelberg. However, the NRMSD is 0.68, which indicates that at OPE the RMSD is almost 70% of the average ΔffCO₂ afternoon signal during Sept. 2020 and Mar. 2021. We perform a





similar synthetic data experiment as for Heidelberg (see Sect. 3.1.2) to investigate, which share of the RMSD can be attributed
to the uncertainty of the observations and which part is due to the (neglected) spatiotemporal variability of the ratios. The
comparison of the synthetic $\Delta CO$-based and $^{14}C$-based $\Delta ffCO_2$ data leads to a RMSD of $1.61 \pm 0.16$ ppm, which already
exceeds the observed RMSD of 1.49 ppm between the observed $\Delta CO$ and $^{14}C$-based $\Delta ffCO_2$. The $\Delta CO$ and $^{14}C$-based $\Delta ffCO_2$
uncertainties can thus fully explain the observed RMSD and the spatiotemporal variability of the ratios in the footprint of the
OPE site seems to have only secondary influence.


Finally, Fig. 5b shows the simulated $\Delta CO$ and $\Delta ffCO_2$ contributions for the flask sampling hours at OPE. A linear regression
through the data yields an average ratio of $8.18 \pm 0.24$ ppb/ppm with high correlation ($R^2$=0.93). There is only a very small
difference <5% between the average ratio of the situations with point source contributions lower than 50% (blue points) and
the very few events with point source contributions larger than 50% (red points). This indicates that the simulations do not
show events with purely point source dominated contributions at OPE, which is in agreement with the observations. However,
the ratio estimated from the model results is 29% lower compared to the ratio from the observations. In contrast, the area
source emissions alone would lead to an average ratio of $10.98 \pm 0.53$ ppb/ppm, which is well in the uncertainty range of the
observed ratio. This could indicate that the contributions from point sources are still overestimated by STILT and/or that also
the emission ratio of the area sources in the footprint of the OPE site are underestimated by TNO. Furthermore, there might be
additional non-fossil CO sources in the footprint of the station, such as biomass burning, which were ignored in STILT.

## 4 Discussion

### 4.1 How large is the uncertainty of an hourly $\Delta CO$-based $\Delta ffCO_2$ estimation based on flask observations?

Vogel et al. (2010) estimated $\Delta CO$-based $\Delta ffCO_2$ at the Heidelberg observation site, using $\Delta CO/\Delta ffCO_2$ ratios from weekly
integrated $^{14}CO_2$ samples. Since the weekly ratios are weighted by the $\Delta ffCO_2$ excess, the $\Delta CO$-based $\Delta ffCO_2$ estimation is
biased towards hours with high $\Delta ffCO_2$. Therefore, Vogel et al. (2010) used flasks to sample the diurnal cycles in summer and
winter, in order to correct the weekly averaged ratios with the diurnal variations. This diurnal cycle correction allowed them
to reduce some of the bias in the $\Delta CO$-based $\Delta ffCO_2$ estimates. In the present study we follow up their weekly integrated
samples based estimation using recently collected hourly based flask samples. Our aim is indeed to investigate whether flask
samples collected with higher frequency and higher temporal resolution can be used to estimate a continuous $\Delta CO$-based
$\Delta ffCO_2$ record and assess the related uncertainty. We used results from hourly flask samples at two contrasting sites, the urban
Heidelberg, and the rural OPE stations.

### 4.1.1 Results from the urban site Heidelberg

In Heidelberg, almost 350 $^{14}CO_2$ flask samples were collected during very different situations between 2019 and 2020. Their
$\Delta CO$ and $\Delta ffCO_2$ excess concentrations compared to the marine background from MHD show a strong correlation with an $R^2$



value of 0.88 (see Fig. 3a). This indicates that the emission ratios of the traffic and heating sectors, which dominate the urban emissions, are quite similar in the main footprint of Heidelberg and the investigated period of time. Furthermore, it follows that the Heidelberg observation site with an air intake height of 30 m above ground is hardly influenced by plumes from nearby point sources, which are associated with rather low emission ratios because large combustion units like power plants have a high combustion efficiency (Dellaert et al., 2019). Indeed, there are very small differences below 3% between the mean afternoon and non-afternoon ratios, and between the average ratios in 2019 and 2020. Moreover, there is almost no seasonal cycle in the ratios, since the average ratio of the flasks collected in the summer half-year is ca. 5% smaller than the average ratio of the flasks from the winter half-year. However, there is a seasonal cycle of the ratios robustness as underlined by the contrasting $R^2$ values between winter and summer.

We must thus emphasize the difficulty to estimate reliable ratios for the summer period and even question the meaning of such an approach. If for example only the flasks from the three main summer months June, July and August are considered, the correlation between $\Delta CO$ and $\Delta ffCO_2$ disappears ($R^2 = 0.06$), which prohibits calculating average summer ratios (see Appendix A1). This has also been found by other studies (Vogel et al., 2010; Miller et al., 2012; Turnbull et al., 2015; Wenger et al., 2019) and can be explained by smaller $\Delta ffCO_2$ signals with large relative uncertainties (see Appendix A1) and/or by the increased contribution from biospheric fluxes and/or non-fossil CO sources during summer (Vimont et al., 2019). Nevertheless, we estimated the average ratio from summer and winter flasks and neglected a potential seasonal cycle in the ratios. However, the global regression line fitting is mostly dominated by the flasks with large $\Delta CO$ and $\Delta ffCO_2$ concentrations, which were predominantly collected during synoptic events in the winter half-year.

The comparison of the $\Delta CO$-based $\Delta ffCO_2$ estimates with the $^{14}C$-based $\Delta ffCO_2$ data from the flasks gives a RMSD of about 4 ppm, which we use as an estimate for the $1\sigma$ uncertainty of the $\Delta CO$-based $\Delta ffCO_2$ concentrations. One half of this uncertainty could be attributed to the measurement uncertainty and the representativeness uncertainty of the CO and $\Delta^{14}CO_2$ background from the marine site MHD. The other half of this uncertainty is related to the ratio variability in the main footprint of Heidelberg, which has been ignored by applying a constant ratio to calculate the $\Delta CO$-based $\Delta ffCO_2$ concentrations. Overall, this uncertainty is almost 4 times larger than the typical uncertainty of $^{14}C$-based $\Delta ffCO_2$ estimates and corresponds to ca. 40% of the mean $\Delta ffCO_2$ signal of the flasks collected in Heidelberg. However, by using the average ratio from the flasks we got a bias-free $\Delta CO$-based $\Delta ffCO_2$ record with hourly resolution. In the companion paper (Maier et al., 2023a) we investigate which observation is better suited to estimate the $ffCO_2$ emissions in the main footprint of Heidelberg – discrete $^{14}C$-based $\Delta ffCO_2$ from flasks with a small uncertainty or continuous $\Delta CO$-based $\Delta ffCO_2$ with hourly coverage but a 4 times larger uncertainty.

### 4.1.2 Results from the rural site OPE

At OPE afternoon $^{14}C$ flasks were collected nearly every 3rd day between September 2020 and March 2021. Since this does not cover a full year, it is not possible to investigate the potential diurnal or seasonal cycle in the ratios. As in the case of





Heidelberg, a constant ratio was used to compute the $\Delta$CO-based $\Delta$ffCO$_2$ record. The flask $\Delta$CO and $^{14}$C-based $\Delta$ffCO$_2$ excess show a lower correlation ($R^2$=0.7) compared to the flasks from the urban site Heidelberg ($R^2$=0.88). This might be explained

by the almost 80% lower mean signal of the flasks collected at OPE and the smaller number of flask samples. This affects the uncertainty of the slope of the regression line, which is at OPE with 0.81 ppb/ppm more than 10 times larger than the one in Heidelberg. The RMSD between the $\Delta$CO-based $\Delta$ffCO$_2$ and the $^{14}$C-based $\Delta$ffCO$_2$ from the flasks is 1.5 ppm, which accounts for almost 70% of the mean $\Delta$ffCO$_2$ signal from the flasks. Compared to the typical $^{14}$C-based $\Delta$ffCO$_2$ uncertainty, the uncertainty of the $\Delta$CO-based $\Delta$ffCO$_2$ is only about 30% higher. This could be explained by the fact that we only considered

the cold period at OPE and that this rural site might be less influenced by the ratio variability. We determined that the whole RMSD of 1.5 ppm can entirely be explained by the measurement uncertainties and the representativeness uncertainty of the background concentrations. Such low ratio variability is expected at more remote sites like OPE as air masses have a long-range transport history with mixing and smoothing of various surface sources. Therefore, this $\Delta$CO-based $\Delta$ffCO$_2$ record with continuous data coverage, if well calibrated with $^{14}$CO$_2$ measurements, could be a valuable addition to discrete $^{14}$C-based

$\Delta$ffCO$_2$ estimates for constraining ffCO$_2$ emissions for afternoon situations during the winter period.

### 4.2 How many $^{14}$CO$_2$ flask measurements are needed to estimate a reliable continuous $\Delta$CO-based $\Delta$ffCO$_2$ record?

We use the STILT forward runs to assess the representativeness of the collected flask samples for the entire period covered by the $\Delta$CO-based $\Delta$ffCO$_2$ record. We compute the average STILT $\Delta$CO/$\Delta$ffCO$_2$ ratios by fitting a regression line through the simulated $\Delta$CO and $\Delta$ffCO$_2$ data for (1) the hours with flask samples only and for (2) all hours covered by the $\Delta$CO-based

$\Delta$ffCO$_2$ record. As the STILT results suggest an unrealistic simulation of situations with more than 50% point source influence in Heidelberg, we restricted the analysis to the hours where STILT predicts a point source influence below 50%. Note that this is by far the largest pool of data (see Fig 4). For Heidelberg, this comparison gives a difference smaller than 3% between the average modelled ratio of the hours with flask sampling events and the average modelled ratio of all hours between 2019 and 2020. This result suggests that the Heidelberg flasks are quite representative for these two years. In the case of OPE, the STILT

average ratio of the hours with flask samplings differs by less than 1% from the average ratio of all afternoon hours between Sept. 2020 and Mar. 2021, indicating again that the flask samples are very representative for the afternoons during this period. Interestingly, STILT suggests a small diurnal cycle in the OPE ratios with an 8% difference between the mean ratios of the afternoon and the non-afternoon hours, respectively.



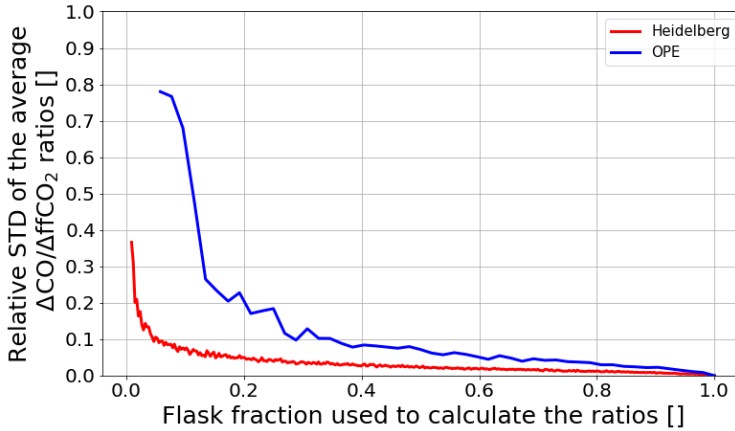

**Figure 6: Results of the bootstrapping experiment. We used an increasing number of random flasks from the Heidelberg (in red) and OPE (in blue) flask pools to deduce an average $\Delta CO/\Delta ffCO_2$ ratio. For each number of flasks, we repeat this experiment 100 times. Finally, we calculate the standard deviation of the average ratios over the 100 repetitions for each number of flasks. Here, we show the relative standard deviation (STD) of the average ratios for an increasing flask fraction used to calculate the ratios. A flask fraction of 1 means that all available flask samples from Heidelberg and OPE, respectively, were used to calculate the average ratios. Obviously, this leads to a standard deviation of 0. See the text for a detailed description of the bootstrapping experiment.**

After having shown that the flask pools from both observation sites seem to be quite representative, we investigate how many flasks are needed to determine a robust average ratio for constructing the $\Delta CO$-based $\Delta ffCO_2$ record. For this, we perform a small bootstrapping experiment. We select from the Heidelberg (and OPE) flask pool randomly i flasks, with i ranging from 3 to the total number of flasks $N_{tot}$ ($N_{tot}$ = 343 flasks in Heidelberg and $N_{tot}$ = 52 flasks at OPE). Then we calculate from the $\Delta CO$ and $\Delta ffCO_2$ data of the i flasks an average ratio $<R_{i,j}>$ by using the regression algorithm from Wurm (2022). We repeat this experiment j=100 times for each i. After that, we can calculate for each i the standard deviation $\sigma(<R_i>)$ over the 100 realizations of $\{<R_{i,1}>,\ldots, <R_{i,100}>\}$. Obviously, we get for i=$N_{tot}$ the average flask ratio $<R_{flask}>$ and $\sigma(<R_{i=N\_tot}>)$=0, as we used all available flasks. Figure 6 shows the relative standard deviation $\sigma(<R_i>)/<R_{flask}>$ for different shares i/$N_{tot}$ of flasks used to calculate the ratio. Apparently, this relative standard deviation of the ratio increases for a decreasing number of flasks used to calculate the ratio. At the urban site Heidelberg, we would need 15 flasks, which are less than 5% of our flask pool, to keep the standard deviation of the ratio below 10%. At the more remote site OPE, we would need 20 flasks, i.e. almost 40% of the collected OPE flasks, to reduce the standard deviation of the ratio to 10%.

Overall, this experiment shows that the number of flasks needed to determine a robust average $\Delta CO/\Delta ffCO_2$ ratio with an uncertainty below 10% depends on the correlation between the $\Delta CO$ and $\Delta ffCO_2$ data. For $R^2$ values between 0.7 and 0.9 it takes about 15 to 20 flasks to determine the average $\Delta CO/\Delta ffCO_2$ ratio with an uncertainty of less than 10%. For this, however, these flasks must cover a wide range of the observed $\Delta CO$ and $\Delta ffCO_2$ concentrations. As mentioned above, the determination of an average ratio is associated with much larger uncertainties during summer with typically lower $R^2$ values. Thus, in order to investigate a potential seasonal cycle in the ratios, it is important to also collect flasks during summer situations with large $\Delta CO$ concentrations. This might increase the chance of getting better correlations and thus lower uncertainties in the summer



ratios. As we considered only one urban and one rural station in this study, we recommend to repeat this experiment at further sites to confirm general applicability.

### 4.3 Can inventory-based $\Delta CO/\Delta ffCO_2$ ratios be used to construct the $\Delta CO$-based $\Delta ffCO_2$ record?

Flask-based $\Delta CO/\Delta ffCO_2$ ratios are independent station-based estimations not influenced by sector specific inventory emission
factors and transport model uncertainties. Moreover, they intrinsically include all potential CO contributions from natural and anthropogenic sources and sinks. However, for many observation sites with continuous CO measurements but without $^{14}$C measurements, the use of inventory-based $\Delta CO/\Delta ffCO_2$ ratios is the only option to estimate hourly $\Delta CO$-based $\Delta ffCO_2$ estimates. Therefore, we also compared the flask-based ratios from Heidelberg and OPE with $\Delta CO/\Delta ffCO_2$ ratios from the TNO inventory transported with STILT to those observational stations.


At the urban site Heidelberg, the model-based estimations face two issues. First, the model predicts events with pure point source emissions which have very low $\Delta CO/\Delta ffCO_2$ ratios of about 2 ppb/ppm, but are hardly observed at the observation site. This illustrates the deficits of STILT to correctly simulate the contributions from point source emissions. Thus, even the improved STILT-VSI approach, which considers the effective emission heights of the point sources seems to overestimate the
contributions from point sources for individual hours. Second, the contributions from the area source emissions alone would lead to an average ratio that is almost 30% lower compared to the average flask ratio. Rosendahl (2022) found during a field campaign next to a highway in Heidelberg, that the measured traffic ratios are about 80% higher compared to the TNO emission ratio for the highway traffic sector. If we assume that the *overall* TNO traffic ratio is underestimated throughout the Rhine Valley and would increase it by 80% in this domain, the modelled ratio of the total area source emissions would increase by
more than 20% and be much closer to the observed flask ratio.

Also, the ratios of the heating sector come along with large uncertainties. In particular, the share of wood combustion has a major impact on the $\Delta CO/\Delta ffCO_2$ ratios of the total heating sector since it releases no $ffCO_2$ emissions but substantial CO emissions. In TNO, the proximity to forested areas (access to wood) is used as a proxy to determine the share of wood
combustion within a grid cell (Kuenen et al., 2022). During two measurement campaigns in two villages around Heidelberg, Rosendahl (2022) showed that this also can lead to biases between the measured and inventory-based heating ratios. Overall, it seems that the TNO emission inventory underestimates the $\Delta CO/\Delta ffCO_2$ ratios in the Rhine Valley during the two years 2019 and 2020. Thus, using those inventory-based (area source) emission ratios would result in strong biases in the order of 40% in the $\Delta CO$-based $\Delta ffCO_2$ estimates.


At the more remote site OPE, the model results show no distinct point source events. This is expected, since the ICOS atmosphere stations are typically located at distances larger than 40 km from large point sources (ICOS RI, 2020). The average simulated $\Delta CO/\Delta ffCO_2$ ratio at OPE turned out to be 30% smaller compared to the average flask ratio. However, if only the



contributions from area sources were considered, the modelled ratio would agree with the flask ratio within their 1σ uncertainty

ranges. We identified three main explanations that could account for the 30% difference between the modelled and observed average ratio. First, the STILT model might overestimate the contributions from the point sources as in Heidelberg. Second, the TNO inventory could underestimate the emission ratios of the area sources, e.g. by an underestimation of the contribution from wood combustion. Chemical characterizations of PM10 highlighted the relative contribution of wood combustion vs. that of fossil fuel in the particulate matter sampled at the station (Borlazza et al., 2022). Third, there is an additional CO contribution

from non-fossil sources, which we ignored in STILT as we only transport the TNO emissions to the observation site.

To investigate the potential contribution from non-fossil CO sources, we calculate the linear regression through the flask $\Delta CO$ and $\Delta ffCO_2$ concentrations by not forcing the regression line through the origin. This yields a slightly larger slope of 11.72 ± 1.09 ppb/ppm with an almost vanishing $\Delta CO$-offset of -1 ± 3 ppb. In principle, this $\Delta CO$-offset could be explained by a

representativeness bias of the MHD CO background or by non-fossil CO contributions between the MHD background site and the OPE observation site. Thus, from this small (and even slightly negative) $\Delta CO$-offset there is no observational evidence for significant non-fossil CO sources or an inappropriate CO background. The former can be confirmed by the top-down inversion results from Worden et al. (2019), who used the Measurements Of Pollution In The Troposphere (MOPITT) CO satellite retrievals in combination with the global chemical transport model GEOS-Chem to calculate monthly gridded (5° x 4°) a-

posteriori CO fluxes for the years 2001 until 2015. The CO fluxes are separated into the three primary source sectors: anthropogenic fossil fuel and biofuel, biomass burning, and oxidation from biogenic non-methane VOCs (NMVOCs). When averaging their results over the 15 years from 2001 until 2015 for the 7 months between September and March, one gets a mean top-down biogenic CO flux of 1.38 nmol/(m$^2$s) in the 4° x 5° grid cell around the OPE site. If we apply this biogenic CO source for the whole European STILT domain, the modelled average $\Delta CO/\Delta ffCO_2$ ratio would only slightly increase from 7.6

± 0.3 ppb/ppm with $\Delta CO$-offset of 3 ± 1 ppb to 7.8 ± 0.4 ppb/ppm with $\Delta CO$-offset of 9 ± 1 ppb. Thus, this non-fossil CO source would mainly affect the $\Delta CO$-offset and might be neglectable during winter. Indeed, the 2001-2015 mean top-down biogenic CO flux in the grid cell around OPE is for the period September to March almost 10 times smaller than the respective anthropogenic CO flux from Worden et al. (2019).

Therefore, we expect that the differences between the modelled and observed average ratio at OPE are rather caused by inconsistencies in the TNO emission ratios or deficits in the transport model. However, for the period April to August, the mean biogenic and mean anthropogenic CO fluxes from Worden et al. (2019) are of the same magnitude, which indicates that the biogenic influence on the $\Delta CO/\Delta ffCO_2$ ratios is much more important during summer than during winter. Overall, these results show that at both sites, the urban Heidelberg site and the rural OPE site, only observation-based ratios should be used

for constructing a continuous $\Delta CO$-based $\Delta ffCO_2$ record. In general, the ratios from emission inventories should be validated by observations if they are to be used to construct a $\Delta CO$-based $\Delta ffCO_2$ record. Otherwise, there could be large biases in the




$\Delta CO$-based $\Delta ffCO_2$ estimates. While the contribution of non-fossil CO sources and sinks in winter seems negligible even at remote stations, additional modeling of the natural CO contributions in summer may be needed, especially for remote sites.

## 5 Conclusions

In the present study, we investigated if [14]C-based $\Delta CO/\Delta ffCO_2$ ratios from flasks collected at the urban site Heidelberg and at the more remote site OPE can be used to construct a continuous $\Delta CO$-based $\Delta ffCO_2$ record for these sites. The almost 350 Heidelberg flasks were sampled during very different meteorological conditions between 2019 and 2020 but show a strong correlation, suggesting similar heating and traffic emission ratios in the Upper Rhine Valley. This average flask $\Delta CO/\Delta ffCO_2$ ratio can thus be used to construct an hourly $\Delta CO$-based $\Delta ffCO_2$ record. The comparison between the $\Delta CO$-based and [14]C-

based $\Delta ffCO_2$ from flasks gives a RMSD of about 4 ppm, which is almost 4 times larger than the typical uncertainty for [14]C-based $\Delta ffCO_2$ estimates. One half of this RMSD is due to observational uncertainties and the other half is caused by the variability of the ratios, which was neglected when applying a constant flask ratio. In a companion paper (Maier et al., 2023a) we further investigate the usage of [14]C- and $\Delta CO$-based $\Delta ffCO_2$ observations to estimate the $ffCO_2$ emissions in the urban surrounding of Heidelberg.


At the rural site OPE, about 50 afternoon flasks were collected from September 2020 to March 2021. Compared to Heidelberg, these flasks show a slightly smaller correlation, but still allowed the determination of a (constant) ratio to construct the $\Delta CO$-based $\Delta ffCO_2$ record for the afternoon hours. The RMSD between $\Delta CO$-based and [14]C-based $\Delta ffCO_2$ from the flasks is about 1.5 ppm, which is about 70% of the mean $\Delta ffCO_2$ signal of the flasks but only about 30% higher than the uncertainty of the

[14]C-based $\Delta ffCO_2$ estimates. At OPE, the RMSD can fully be explained by the observational uncertainties alone, which indicates that atmospheric transport has smoothed out the spatiotemporal variability of the emission ratios. Therefore, it is interesting to investigate if the continuous $\Delta CO$-based $\Delta ffCO_2$ record could provide additional spatiotemporal information to constrain the $ffCO_2$ emissions around a remote site.

Overall, this study highlights a number of challenges and limitations in estimating $\Delta CO$-based $\Delta ffCO_2$ concentrations for an urban and a remote site. Urban sites like Heidelberg with large CO and $ffCO_2$ signals allow the estimation of $\Delta CO/\Delta ffCO_2$ ratios with typically smaller uncertainties. However, the spatiotemporal variability of the ratios from nearby emissions has a strong impact on the overall $\Delta CO$-based $\Delta ffCO_2$ uncertainty. In contrast, the heterogeneity in the fossil emission ratios seems to be smoothed out at remote sites like OPE. However, at these sites it is more difficult to calculate average ratios due to the

lower correlations between $\Delta CO$ and [14]C-based $\Delta ffCO_2$, which might be caused by the smaller signals and a relatively larger influence from non-fossil CO sources and sinks, especially during summer.





Finally, we also compared the flask-based ratios with simulated ratios using the TNO inventory and the STILT transport model. At both sites there are significant differences between the observed and the modelled ratios, which might be caused by
inconsistencies in the TNO emission ratios and deficits in the STILT transport model. Consequently, using the inventory-based estimated ratio would lead to systematic biases in the $\Delta$CO-based $\Delta$ffCO$_2$ record. We also assessed how many flasks are needed to estimate a robust ratio that could be applicable to derive $\Delta$CO-based $\Delta$ffCO$_2$ at hourly resolution. Our results suggests that about 15 to 20 flasks could be used to determine the average $\Delta$CO/$\Delta$ffCO$_2$ ratio with an uncertainty of less than 10% for the winter period. Overall, our results illustrate the importance of maintaining and developing the radiocarbon observation network
to validate the sector-specific bottom-up CO/ffCO$_2$ emission ratios. They also suggest that campaign-based validation using a traveling flask sampler could be valuable for estimating the ratios at stations where CO measurements are performed without $^{14}$C measurements.










## Appendix

### A1. How to estimate the average $<\Delta CO/\Delta ffCO_2>$ ratio from error-prone $\Delta CO$ and $\Delta ffCO_2$ observations

Here, we show why one should use a weighted total least squares regression to calculate average $<\Delta CO/\Delta ffCO_2>$ ratios from error-prone $\Delta CO$ and $\Delta ffCO_2$ observations. For this, we perform a synthetic data experiment. We use the *positive* $^{14}C$-based

$\Delta ffCO_2$ concentrations from the Heidelberg flasks as the synthetic "true", i.e., error-free $\Delta ffCO_2$ observations and multiply them with a constant "true" $\Delta CO/\Delta ffCO_2$ ratio of 8.44 ppb/ppm to get synthetic "true" $\Delta CO$ observations (see Fig. A1a). We then draw random numbers from an unbiased Gaussian distribution with $1\sigma$ range of 1.16 ppm (for $\Delta ffCO_2$) and 14.49 ppb (for $\Delta CO$), which corresponds to the mean uncertainties of the real flask observations. We add those random numbers to the "true" $\Delta ffCO_2$ and $\Delta CO$ concentrations, respectively, to get synthetic error-prone data (see Fig. A1b). If we plot the synthetic

error-prone $\Delta CO/\Delta ffCO_2$ ratios against the synthetic error-prone $\Delta ffCO_2$ concentrations, we get a large scattering for low $\Delta ffCO_2$ concentrations (see Fig. A1c). This scattering is only caused by the uncertainties as we have assumed a constant ratio in this synthetic data experiment.

For a comparison, we now can calculate the arithmetic mean, the error-weighted mean, and the median of the synthetic error-

prone ratios, as well as the slope of a weighted total least squares regression line from Wurm (2022) through the synthetic error-prone $\Delta CO$ and $\Delta ffCO_2$ data. To get better statistics we repeat this experiment 10'000 times. On average, we get the following results (average ± standard deviation over 10'000 repetitions):

- Arithmetic mean of the ratios: 9.42 ± 77.84 ppb/ppm
- Error-weighted mean of the ratios: 8.24 ± 0.08 ppb/ppm

- Median of the ratios: 8.39 ± 0.11 ppb/ppm
- Slope of regression line: 8.44 ± 0.06 ppb/ppm

This indicates that only the slope of a regression line, which takes into account the uncertainty of the $\Delta CO$ and $\Delta ffCO_2$ data yields the initial "true" constant ratio of 8.44 ppb/ppm. The arithmetic mean of the ratios shows the largest deviation to the "true" ratio with a very large variability within the 10'000 repetitions. This can be explained by the widely scattering ratios

during situations with low $\Delta ffCO_2$ concentrations but huge relative $\Delta ffCO_2$ uncertainties. The error-weighted mean ratio and the median ratio is on average 2.4% and 0.6%, respectively, too low. This bias might be introduced by negative ratios, which are caused by very small synthetic "true" $\Delta CO$ or $\Delta ffCO_2$ data that became negative after adding the random uncertainty contribution. Therefore, we recommend to use a weighted total least squares algorithm to calculate the average $<\Delta CO/\Delta ffCO_2>$ ratio.





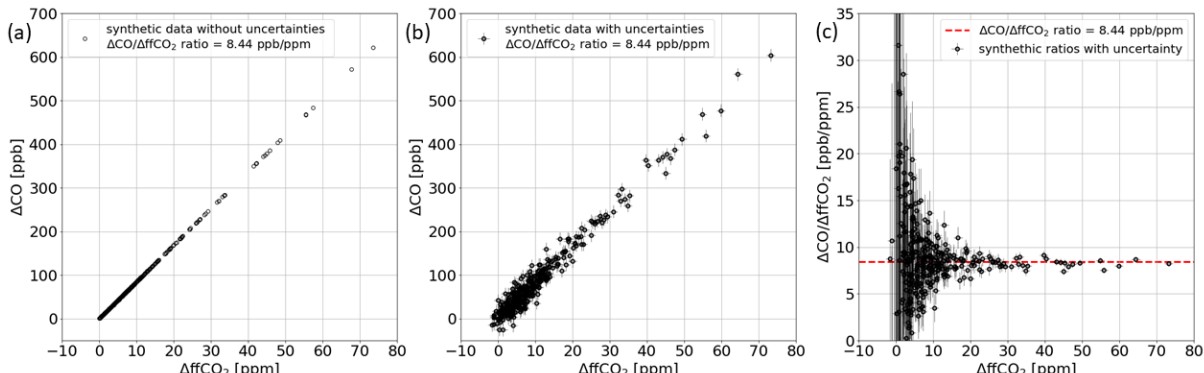

**Figure A1: (a) synthetic "true" ΔCO and ΔffCO₂ data with a constant ratio of 8.44 ppb/ppm. (b) synthetic error-prone ΔCO and ΔffCO₂ data under the assumption of a constant ratio of 8.44 ppb/ppm. (c) synthetic error-prone ΔCO/ΔffCO₂ ratios.**

This synthetic data experiment simulates the situation at an urban site like Heidelberg with a large range of ΔCO and ΔffCO₂ concentrations. In this case, we have a very good correlation between the ΔCO and ΔffCO₂ data. Indeed, the $R^2$-value from the applied regression is on average $0.968 \pm 0.003$ and the uncertainty of the slope is on average 0.06 ppb/ppm. But what happens if we have a smaller range of ΔCO and ΔffCO₂ data, for example at a remote site or during summer? To answer this, we perform the synthetic data experiment again, but only with synthetic "true" ΔffCO₂ concentrations that are smaller than 5 ppm. This increases the uncertainty of the slope to 0.55 ppb/ppm, which is almost a factor of 10. Moreover, the $R^2$-value dramatically decreases to $0.08 \pm 0.12$. This shows the difficulty of calculating average ratios during summer or at very remote sites with low ΔffCO₂ signals (even in the absence of non-fossil CO sources).

In Sect. 3.1.2 we want to estimate the contribution of the observational uncertainties (i.e., the measurement and background representativeness uncertainty) to the RMSD between the ΔCO- and $^{14}$C-based ΔffCO₂ concentrations of the Heidelberg flasks. For this, we used the average flask <ΔCO/ΔffCO₂> ratio to calculate from the error-prone ΔCO data (see Fig. A1b) synthetic ΔCO-based ΔffCO₂ concentrations. In Fig. 3b (red and orange dots for winter and summer flasks), we plot these synthetic ΔCO-based ΔffCO₂ data against the error-prone synthetic ΔffCO₂ concentrations.







## A2. Summer vs. winter ratios

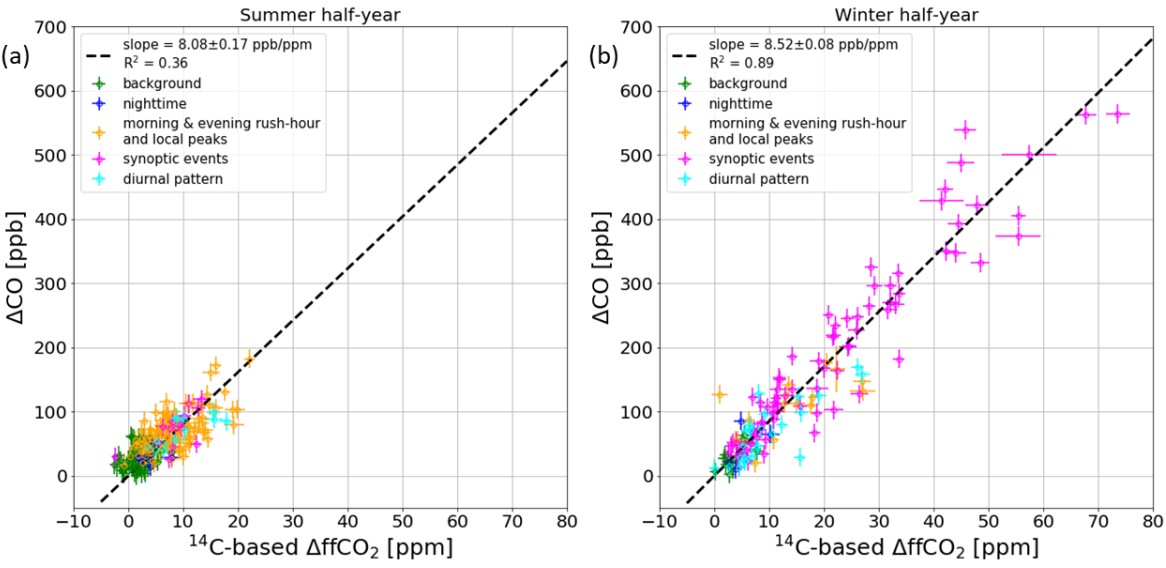

**Figure A2:** Scatter plot with the measured ΔCO and the $^{14}$C-based ΔffCO$_2$ concentrations of the hourly flasks collected at the Heidelberg observation site between 2019 and 2020 during (a) the summer half-year and (b) the winter half-year. The colors indicate the sampling situation of the flasks (see description in the caption of Fig. 2). The black dashed line shows a regression line performed with the weighted total least squares algorithm from Wurm (2022).

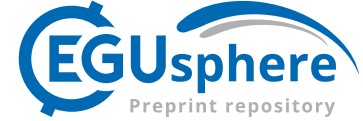

**Data availability**

Data will be made available at the Heidelberg University data depository.

**Author contribution**

FM designed the study together with IL, SH and MG. SH and SC provided the observations from Heidelberg and OPE, respectively. HDvdG was responsible for the TNO emission inventory. FM evaluated the data and conducted the modelling. FM wrote the manuscript with help of all co-authors.

**Competing interests**

The authors declare that they have no conflict of interest.

**Acknowledgement**

We thank Julian Della Coletta, Sabine Kühr, Eva Gier, and the whole staff of the ICOS Central Radiocarbon Laboratory (CRL) for conducting the continuous measurements in Heidelberg and preparing the $^{14}CO_2$ analyses. Moreover, we like to thank Ronny Friedrich from the Curt-Engelhorn Center Archaeometrie (CECA), who performed the AMS measurements. We are grateful to the staff of the ICOS site OPE for collecting the $^{14}CO_2$ flask samples and conducting the continuous measurements.

We thank the ICOS Flask and Calibration Laboratory (FCL) for measuring all flask concentrations. Furthermore, we are grateful to the National Oceanic and Atmospheric Administration Global Monitoring Laboratory (NOAA GML) for providing the flask $CO_2$ and CO measurements from Mace Head. We further thank the staff of TNO at the Department of Climate, Air and Sustainability in Utrecht for providing the emission inventory as well as Julia Marshall and Michał Gałkowski, who computed and processed the high-resolution WRF meteorology in the Rhine Valley.

**Financial support**

This research has been supported by the German Weather Service (DWD), the ICOS Research Infrastructure and VERIFY (grant no. 776810, Horizon 2020 Framework). The ICOS Central Radiocarbon Laboratory is funded by the German Federal Ministry of Transport and Digital Infrastructure.




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
