# Peer review of "Uncertainty of continuous $\Delta\text{CO}$ -based $\Delta\text{ffCO}_2$ estimates derived from $^{14}\text{C}$ flask and bottom-up $\Delta\text{CO}/\Delta\text{ffCO}_2$ ratios"

_EGUsphere, 2023_

## Referee Comment (RC1)

The manuscript by Maier et al. presented measurements of $^{14}CO2$ and CO in flasks collected at two sites, and the purpose of these measurements is to assess the ratio and the associated uncertainties of CO and fossil-fuel sourced CO2 (ΔffCO2). The latter, proposed by the authors, can be further used to estimate ΔffCO2 based on CO measurements which is easy doing. The idea behind this study appears to be that since C-14 analysis is difficult and there is a need of alternatives to estimate ΔffCO2, while CO that is produced mainly from the same fossil fuel combustion source can serve as a tracer to estimate ΔffCO2 as long as the ratio of CO to ΔffCO2 can be constrained.

I would say this is a fair theory, however, the manuscript is more like a measurement report rather than a research article. This is my first impression of reading this manuscript. The second, as the authors indicated in the manuscript, there are many issues regarding the ratio based on flask measurements, so as the ability of this ratio used to predict ΔffCO2 from CO measurements. This whole idea has a few confusions or flaws that leads to little confidence for its applications. I list my major concerns as follows:

1). The authors point out the disadvantage of using C-14 measurement to assess ΔffCO2, high cost, low coverage, and etc., and thus there is a need of independent tracers that are easy monitoring. The proposed tracer is CO. But in order to use CO to estimate ΔffCO2 in a site or regions, one has to first measure enough C-14 data to build a robust CO/ΔffCO2 ratio? This is sort of a loop, if there is already C-14 measurements, isn't that can be directly used to calculate ΔffCO2? Although CO measurements can be continuous, so as the derived ΔffCO2 with a known CO/ΔffCO2 ratio, would continuous ΔffCO2 record be offering significantly more valuable information than discrete (e.g., daily or hourly average) ΔffCO2 measurements?

2) I am not sure if this is the authors' idea: once a robust CO/ΔffCO2 ratio was available, then perhaps this ratio is able to apply to other locations or the same location but at a different time? This is my impression of reading the manuscript, and this is perhaps the most (if not only) significance of establishing the ratio. However, as the measurement reports from the two sites indicates, the CO/ΔffCO2 ratio are different among these sites, and there are also large temporal variations. This causes doubts on the ability of the ratio to be applied to period or regions without

C-14 but only CO observations. In fact, due to spatial and temporal variations in the use of fossil fuel energy, as well as the spatial and temporal variations in other sources/factors that could influence the production and atmospheric removal of CO, there is a doubt that whether the CO/$\Delta$ffCO2 ratio can stay relatively constant with time and space.

The first concern emphasizes that no matter what one needs to measure C-14, and the second indicates the established ratio is probably not able to fulfil the intended tasks. As such, the scientific significance of these measurements is somewhat weak, and this is why I suggested that this manuscript is more like a measurement report.

In addition, I have some technical suggestions that the authors can consider to use or not:

1) in the method part, please state more clearly how other sources especially the biomass burning source of CO is treated to get $\Delta$CO; This includes the modeling of inventory based ratio.
2) In 2.2.1., the defined $R_{flask}$ is in fact not used but instead $\Delta$CO/$\Delta$ffCO2 in follow-up results and discussion. I suggest to keep constant throughout the manuscript.
3) Figure 3b, could not understand how real flask measurements can be plotted in the figure, the X-axis is measurements, while the Y-axis is synthetic data which is not scaled to real measurements.

---

## Referee Comment (RC2)

[revised manuscript text omitted]

$$\Delta\text{ffCO}_2^{hrly} = \frac{\Delta\text{CO}^{hrly}}{\langle R_{flask}\rangle} \qquad\qquad (1)$$

To calculate the ΔCO and the $^{14}$C-based ΔffCO₂ excess concentrations at the Heidelberg and OPE observation sites, we must choose an appropriate CO and $\Delta^{14}\text{CO}_2$ background. Back-trajectory analyses by Maier et al. (2023b) show a predominant westerly influence for stations in Central Europe; about 2/3 of all back-trajectories, which were calculated for nine European ICOS sites for the full year 2018, end over the Atlantic Ocean at the western boundary of the European continent. Indeed, we identified Mace Head (MHD, 53.33°N, 9.90°W, 5 m a.s.l.), which is located at the west coast of Ireland, to be an appropriate marine reference site for Central Europe. Therefore, we use smooth fit curves through weekly CO flask results (Petron et al., 2022) and two-week integrated $\Delta^{14}\text{CO}_2$ samples from MHD as our CO and $\Delta^{14}\text{CO}_2$ background, respectively. The applied curve fitting algorithm was developed by the National Oceanic and Atmospheric Administration (NOAA, Thoning et al., 1989). This algorithm yields a fit standard deviation of 13.37 ppb and 0.86‰, respectively, for the CO and $\Delta^{14}\text{CO}_2$ background curves.

Obviously, MHD is a less representative background station for situations with non-western air masses. For this, Maier et al. (2023b) estimated a representativeness bias and uncertainty for the MHD background of 0.09±0.28 ppm ffCO₂ for hourly flask samples collected in Central Europe at the eastern ICOS site Křešín. We assume this estimate to be an upper limit for the Heidelberg and OPE sites, which are located further to the West. Therefore, we decided to neglect the representativeness bias in our calculations. However, we take into account its variability, which is the representativeness uncertainty of the MHD background. The 0.28 ppm ffCO₂ uncertainty would result in a representativeness uncertainty of 0.64‰ for the MHD $\Delta^{14}\text{CO}_2$ background, if one assumes that a 1 ppm ffCO₂ signal is caused by a 2.3‰ $\Delta^{14}\text{CO}_2$ depletion (we deduced this conversion factor from the Heidelberg flask results). Similarly, we can estimate the representativeness uncertainty for the CO background, if we assume a mean CO/ffCO₂ ratio to convert the estimated 0.28 ppm ffCO₂ uncertainty into a CO uncertainty. The TNO inventory suggests for the Eastern boundary of our model domain (within 22-23°E and 37-61°N) a mean CO/ffCO₂ emission ratio of roughly 18 ppb/ppm in 2020. We use this ratio as an upper limit and get a CO background representativeness uncertainty of 0.28 ppm*18 ppb/ppm=5.04 ppb. To estimate the overall CO and $\Delta^{14}\text{CO}_2$ background uncertainty, we add the fit uncertainty and the representativeness uncertainty quadratically, which yields 14.29 ppb and 1.07‰, respectively.

**2.2.1 Calculation of an observation-based <R$_{flask}$> ratio**

To calculate the $^{14}$C-based <R$_{flask}$> ratio, we first estimate the ΔffCO₂ concentrations from the $\Delta^{14}\text{
[revised manuscript text omitted]

---

## Author Comment (AC1)

We want to thank the anonymous referee for the review of our manuscript. Our replies are marked in blue.

The manuscript by Maier et al. presented measurements of 14CO2 and CO in flasks collected at two sites, and the purpose of these measurements is to assess the ratio and the associated uncertainties of CO and fossil-fuel sourced CO2 ($\Delta$ffCO2). The latter, proposed by the authors, can be further used to estimate $\Delta$ffCO2 based on CO measurements which is easy doing. The idea behind this study appears to be that since C-14 analysis is difficult and there is a need of alternatives to estimate $\Delta$ffCO2, while CO that is produced mainly from the same fossil fuel combustion source can serve as a tracer to estimate $\Delta$ffCO2 as long as the ratio of CO to $\Delta$ffCO2 can be constrained.

I would say this is a fair theory, however, the manuscript is more like a measurement report rather than a research article. This is my first impression of reading this manuscript. The second, as the authors indicated in the manuscript, there are many issues regarding the ratio based on flask measurements, so as the ability of this ratio used to predict $\Delta$ffCO2 from CO measurements. This whole idea has a few confusions or flaws that leads to little confidence for its applications. I list my major concerns as follows:

We regret that we have not succeeded in explaining the relevance of our study to the broader community. As we state in the introduction, the basic idea of using CO as a proxy for ffCO$_2$ is more than 20 years old. Even older is the realisation that there will be no semi-continuous measurements of the direct ffCO$_2$ tracer $\Delta^{14}$CO$_2$ in atmospheric observing networks for the foreseeable future. As we discuss in the paper, CO is not a perfect ffCO$_2$ proxy for many reasons. Therefore, we consider it the primary goal of our study to emphasise the shortcomings of the CO-proxy approach clearly. This includes showing that the use of the CO proxy requires calibration by $^{14}$C measurements to achieve the necessary precision. Despite all the difficulties and deficits of the CO-proxy-based ffCO$_2$ estimation, we demonstrate in this paper and in the companion paper by Maier et al. (2023) the great potential of this method. We understand that due to our comprehensive and in-depth analysis of the problems of the CO-proxy, the impression of the measurement report may arise. However, it is precisely this detailed and dedicated investigation of the CO-proxy approach that sets this publication apart from its predecessors and makes future users of the CO-proxy approach, whether for in-situ or remote sensing approaches, aware of its potential and shortcomings.

1). The authors point out the disadvantage of using C-14 measurement to assess $\Delta$ffCO2, high cost, low coverage, and etc., and thus there is a need of independent tracers that are easy monitoring. The proposed tracer is CO. But in order to use CO to estimate $\Delta$ffCO2 in a site or regions, one has to first measure enough C-14 data to build a robust CO/$\Delta$ffCO2 ratio? This is sort of a loop, if there is already C-14 measurements, isn't that can be directly used to calculate $\Delta$ffCO2? Although CO measurements can be continuous, so as the derived $\Delta$ffCO2 with a known CO/$\Delta$ffCO2 ratio, would continuous $\Delta$ffCO2 record be offering significantly more valuable information than discrete (e.g., daily or hourly average) $\Delta$ffCO2 measurements?

As mentioned above, using $^{14}CO_2$ measurements is the most direct way to estimate $\Delta ffCO_2$ concentrations. However, the sparse $^{14}CO_2$ observations lead to a small amount of $\Delta ffCO_2$ estimates, which could be used in inverse models to infer $ffCO_2$ emissions. The question is then: is the amount of $\Delta ffCO_2$ data enough for getting robust and data-driven inversion results? This is exactly what we investigated in the companion paper (Maier et al., 2023), which followed up this study. It turned out, that we indeed get no robust $ffCO_2$ emission estimates for the surroundings of Heidelberg if we use the $^{14}C$-based $\Delta ffCO_2$ estimates from the Heidelberg flasks. This can mainly be explained by the very heterogenic distribution of the $ffCO_2$ sources around Heidelberg (including several large point sources) and the shortcomings of the transport model in accurately simulating $ffCO_2$ concentrations for individual (afternoon) hours. In contrast, the continuous Heidelberg CO-based $\Delta ffCO_2$ estimates, which we derived in this study, yield robust and data-driven inversion results that could be used to investigate the effect of the Corona restrictions and to validate the seasonal cycle of the TNO $ffCO_2$ emission inventory in the main footprint of Heidelberg. Thus, the continuous CO-based $\Delta ffCO_2$ record indeed yields more valuable information than the discrete $^{14}C$-based $\Delta ffCO_2$ estimates from flasks for this urban region around Heidelberg and at the present time when the traffic and heating sectors show here similar $CO/ffCO_2$ emission ratios. Of course, if the $^{14}C$-based $\Delta ffCO_2$ had a similar temporal resolution like the CO-based $\Delta ffCO_2$, the information content of the $^{14}C$-based $\Delta ffCO_2$ would be higher than that of the CO-based $\Delta ffCO_2$. However, as this is not (yet) the case, we conclude that the usage of CO-based $\Delta ffCO_2$ to infer $ffCO_2$ emissions could be a valuable approach for other (urban) sites.
For such a CO-based $\Delta ffCO_2$ inversion, it is essential to have a reliable quantification of the uncertainties of the CO-based $\Delta ffCO_2$ estimates and an investigation of potential seasonal or diurnal cycles in the $\Delta CO/\Delta ffCO_2$ ratios. Both aspects are treated in detail in our present study, which is why it forms the basis for CO-based $\Delta ffCO_2$ inversions.

2) I am not sure if this is the authors' idea: once a robust $CO/\Delta ffCO2$ ratio was available, then perhaps this ratio is able to apply to other locations or the same location but at a different time? This is my impression of reading the manuscript, and this is perhaps the most (if not only) significance of establishing the ratio. However, as the measurement reports from the two sites indicates, the $CO/\Delta ffCO2$ ratio are different among these sites, and there are also large temporal variations. This causes doubts on the ability of the ratio to be applied to period or regions without C-14 but only CO observations. In fact, due to spatial and temporal variations in the use of fossil fuel energy, as well as the spatial and temporal variations in other sources/factors that could influence the production and atmospheric removal of CO, there is a doubt that whether the $CO/\Delta ffCO2$ ratio can stay relatively constant with time and space.

We agree, the $\Delta CO/\Delta ffCO_2$ can show large spatial and temporal differences due to the spatiotemporal variability of the $CO/ffCO_2$ emission ratios and/or non-fossil CO contributions. This is also what we found in our study: the average $\Delta CO/\Delta ffCO_2$ ratio in Heidelberg is lower compared to that at OPE. That's why we do not recommend to apply a single $\Delta CO/\Delta ffCO_2$ ratio for different sites or times, if it is not validated by $^{14}C$ measurements at the respective sites. In our study, we discuss how many $^{14}C$ measurements should be performed to get robust $\Delta CO/\Delta ffCO_2$ ratios, which can then be used to construct a CO-based $\Delta ffCO_2$ record with high temporal resolution.

Moreover, our comparison of the observed ratios with inventory-based ratios from TNO clearly reveals that the $\Delta CO/\Delta ffCO_2$ ratios should be calibrated with $^{14}C$ measurements to avoid large biases in the CO-based $\Delta ffCO_2$ estimates. Ongoing $^{14}C$ measurements are therefore a prerequisite for monitoring changes in future $\Delta CO/\Delta ffCO_2$ ratios and continuing to apply the CO proxy approach. This information is also relevant for studies combining satellite-based CO observations and inventory-based $CO/ffCO_2$ emission ratios to derive $ffCO_2$ emissions.

The first concern emphasizes that no matter what one needs to measure C-14, and the second indicates the established ratio is probably not able to fulfil the intended tasks. As such, the scientific significance of these measurements is somewhat weak, and this is why I suggested that this manuscript is more like a measurement report.

Unfortunately, it is not straightforward to calculate $\Delta ffCO_2$ concentrations, which then could be used to derive top-down $ffCO_2$ emission estimates. If there were continuous $^{14}C$ measurements with high spatiotemporal coverage and low uncertainty, we would hardly need to think about using CO as a tracer for $ffCO_2$. However, this is at least not yet the case. Thus, the CO proxy approach is a way to bridge the gap between more reliable but sparse $^{14}C$-based $\Delta ffCO_2$ estimates with temporal information derived from CO.

With this in mind, we are convinced that our present study is important and has a scientific significance. We want to summarize here the main points, why we do not think that this study is just a measurement report:

1. In addition to quantifying the uncertainties of the CO-based $\Delta ffCO_2$, we conducted a synthetic data experiment to characterize and interpret the cause of these uncertainties. From this study, we were able to draw conclusions about the impact of the spatiotemporal variability of the $CO/ffCO_2$ emission ratios on the CO-based $\Delta ffCO_2$ uncertainties at an urban and a rural site.

2. We used the STILT model to also simulate $\Delta CO/\Delta ffCO_2$ ratios for both sites. We carefully compared the simulated ratios with our measurements and demonstrate that there can be substantial biases. Our study suggests that the TNO inventory may underestimate the $CO/ffCO_2$ emission ratios in the Rhine Valley. A more detailed investigation (in the revised version of the manuscript) reveals that especially the $CO/ffCO_2$ emissions from the TNO heating sector might be too low in the main footprint of Heidelberg. Such studies are important to improve the emission inventories.

3. By performing a bootstrapping approach, we provide a rough estimate on how many $^{14}C$ flasks should be collected at other sites to obtain robust $\Delta CO/\Delta ffCO_2$ ratios, which can then be used to derive CO-based $\Delta ffCO_2$ concentrations at those sites. This is an essential information if CO-based $\Delta ffCO_2$ data is to be used in inverse models to estimate $ffCO_2$ emissions. In our follow-up study, we show that the CO-based $\Delta ffCO_2$ data from Heidelberg lead to robust $ffCO_2$ emission estimates, which could be used to validate the seasonal cycle of the TNO emission inventory.

From that we argue that our study has enough scientific significance to be published as a research article.

In addition, I have some technical suggestions that the authors can consider to use or not:

1) in the method part, please state more clearly how other sources especially the biomass burning source of CO is treated to get ΔCO; This includes the modeling of inventory based ratio.

The TNO inventory includes CO emissions from agricultural waste burning, i.e. the burning of crop residues based on a 10 year climatology. However, CO emissions from forest fires are not included. These emissions are erratic and can be obtained from e.g. the Copernicus Global Fire assimilation system (GFAS). For the central European region, forest fires are not a dominant source of CO. We clarified this in the manuscript. (p. 7, l. 205-207)

2) In 2.2.1., the defined Rflask is in fact not used but instead ΔCO/ΔffCO2 in follow-up results and discussion. I suggest to keep constant throughout the manuscript.

Thank you for this hint. We replaced Rflask by $\Delta CO/\Delta ffCO_2$ in our manuscript.

3) Figure 3b, could not understand how real flask measurements can be plotted in the figure, the X-axis is measurements, while the Y-axis is synthetic data which is not scaled to real measurements.

The real and synthetic data share the same X-axis and Y-axis. We labelled the X-axis with "$^{14}$C-based $\Delta ffCO_2$" because it refers to the $^{14}$C-based $\Delta ffCO_2$ observations (in case of the real observations, i.e. the black and grey dots) and to the error-prone synthetic $\Delta ffCO_2$ data (in case of the synthetic data, i.e. the red and orange dots), which should mimic the variability of the $^{14}$C-based $\Delta ffCO_2$ observations. Similarly, the Y-axis with label "$\Delta CO$-based $\Delta ffCO_2$" refers to the $\Delta CO$-based $\Delta ffCO_2$ observations as well as to the synthetic $\Delta CO$-based $\Delta ffCO_2$ data, which were constructed by dividing the synthetic $\Delta CO$ data by a constant $\Delta CO/\Delta ffCO_2$ ratio of 8.44 ppb/ppm. We tried to make this clearer in the caption of Fig. 3.

References:

Maier, F. M., Rödenbeck, C., Levin, I., Gerbig, C., Gachkivskyi, M., and Hammer, S.: Potential of $^{14}$C-based versus $\Delta CO$-based $\Delta ffCO_2$ observations to estimate urban fossil fuel $CO_2$ (ffCO$_2$) emissions, EGUsphere [preprint], https://doi.org/10.5194/egusphere-2023-1239, 2023.

---

## Author Comment (AC2)

We want to thank John Miller for the review of our manuscript and his helpful suggestions for improving this study. Our replies are marked in blue.

Review of Maier et al, "Uncertainty of continuous $\Delta CO$-based $\Delta ffCO2$ estimates derived from [14]C flask and bottom-up $\Delta CO/\Delta ffCO2$ ratios" by John Miller

**General comments**

Overall, this is a very good study focused on developing high frequency proxies (here CO) for the estimation for the recently added fossil content of atmospheric $CO_2$ measurements. The writing is generally very good and the figures are excellent, reflecting the analysis itself. The pdf has numerous comments but I will highlight two here:

1. Although this is clear in the title, use of "Delta(CO)-based DffCO2 estimates", when used without explanation, has the potential to be highly misleading because these estimates (the atmospheric data-based ones the paper shows to be trustworthy) are still based on D[14]C I'm not exactly sure of the solution, but perhaps you can employ nomenclature/notation that identifies such values as 'calibrated' by [14]C.

   We agree, that this can lead to confusion. We tried to make this clearer in our manuscript, e.g. by using "proxy-based $\Delta ffCO_2$" in the abstract, as you suggested. However, in the main text we would like to use "$\Delta CO$-based $\Delta ffCO_2$" because this is also what we used in the companion paper (Maier et al., 2023). We also added some text in the introduction, to make it clear, that the "$\Delta CO$-based $\Delta ffCO_2$" is calibrated with [14]C. (e.g., p. 3, l. 71-72; p. 3, l. 90; p. 5, l. 137)

2. I have a few questions about the TNO inventory that could benefit by a bit more investigation and explanation. First, it appears that TNO includes biofuels such as wood. But what about ethanol and biodiesel? Generally, can the fossil components of the TNO inventory be isolated for a more direct comparison with [14]C-based observations? Second, in investigation of the point source impacts for Heidelberg, can you transport the non-point-source sectors to see how much the match to data is improved – i.e., is the mismatch mainly due to the ratio of (dilute) point to area sources in TNO or mainly due to incorrect emission ratios for the area sources?

   Thank you for your suggestions. The TNO inventory distinguishes between fossil fuel and biofuel $CO_2$ and CO emissions. Thus, TNO includes emissions from wood-fired heating as well as biofuel emissions from the traffic sector. For the traffic sector this is based on national reporting of shares of ethanol in gasoline and/or biodiesel in diesel. Moreover, there is also data on the amount of biomass co-fired in coal-fired powerplants. This is also in the inventory as CO2 from biofuel.

We carried out both of the simulations you suggested (see Fig. 1 below). The magenta dots in Figure 1a show the contributions from the non-point-source sector only. They lead to an average ratio of about 6 ppb/ppm, which is well below the observed ratio (grey crosses). From that we conclude that TNO might underestimate the ratio of the area source emissions in the Rhine Valley. Furthermore, a potential wrong dilution ratio between point sources and area sources (both with correct emission ratios) can't explain the differences between observed and modelled $\Delta CO/\Delta ffCO_2$ ratios, because both, the average modelled ratio from only-area sources (6.0 ppb/ppm) and the ratio from only-point sources (1.2 ppb/ppm) are below the observed ratio of 8.44 ppb/ppm.

Fig. 1b shows the simulated contributions from the traffic (orange) and heating (cyan) sectors. The traffic (biofuel plus fossil fuel) sector leads to an average $\Delta CO/\Delta ffCO_2$ ratio of 7.72±0.08 ppb/ppm ($R^2$=0.93), which is less than 10% lower compared to the observed average ratio of 8.44 ppb/ppm. However, the heating (wood plus fossil) sector leads to a much lower average $\Delta CO/\Delta ffCO_2$ ratio of 3.36±0.09 ppb/ppm ($R^2$=0.65). This might indicate that the $CO/ffCO_2$ emission ratios from the TNO heating sector are too low in the main footprint of Heidelberg. This could be explained, for example, by an incorrect distribution of the use of fossil and bio fuels in the heating sector.

We included these plots and a discussion in our manuscript. (Fig. 3b and A3 in the manuscript; p. 12, l. 313-314; p. 18-19, l. 474-499)

[Figure]

*Figure 1: $\Delta CO$ and $^{14}C$-based $\Delta ffCO_2$ observations from the Heidelberg flasks (in grey) and (a) simulated non-point-source $\Delta CO$ and $\Delta ffCO_2$ contributions for the flask events (magenta), and (b) simulated $\Delta CO$ and $\Delta ffCO_2$ contributions from the traffic (orange) and heating (cyan) sectors for the flask events.*

**Specific comments**

Suggested edits and comments are embedded in the manuscript .pdf. Blue highlights indicate those that are language oriented and yellow for science/conceptual issues.

Thank you for your comments. We directly replied to them in the .pdf document.

References:

Maier, F. M., Rödenbeck, C., Levin, I., Gerbig, C., Gachkivskyi, M., and Hammer, S.: Potential of $^{14}$C-based versus $\Delta$CO-based $\Delta$ffCO$_2$ observations to estimate urban fossil fuel CO$_2$ (ffCO$_2$) emissions, EGUsphere [preprint], https://doi.org/10.5194/egusphere-2023-1239, 2023.

---

## Author Comment (AC3)

[revised manuscript text omitted]

---

## Author Response (AR2)

**Replies to review 1**

We want to thank the anonymous referee for the review of our manuscript. Our replies are marked in blue.

The manuscript by Maier et al. presented measurements of 14CO2 and CO in flasks collected at two sites, and the purpose of these measurements is to assess the ratio and the associated uncertainties of CO and fossil-fuel sourced CO2 (ΔffCO2). The latter, proposed by the authors, can be further used to estimate ΔffCO2 based on CO measurements which is easy doing. The idea behind this study appears to be that since C-14 analysis is difficult and there is a need of alternatives to estimate ΔffCO2, while CO that is produced mainly from the same fossil fuel combustion source can serve as a tracer to estimate ΔffCO2 as long as the ratio of CO to ΔffCO2 can be constrained.

I would say this is a fair theory, however, the manuscript is more like a measurement report rather than a research article. This is my first impression of reading this manuscript. The second, as the authors indicated in the manuscript, there are many issues regarding the ratio based on flask measurements, so as the ability of this ratio used to predict ΔffCO2 from CO measurements. This whole idea has a few confusions or flaws that leads to little confidence for its applications. I list my major concerns as follows:

We regret that we have not succeeded in explaining the relevance of our study to the broader community. As we state in the introduction, the basic idea of using CO as a proxy for ffCO₂ is more than 20 years old. Even older is the realisation that there will be no semi-continuous measurements of the direct ffCO₂ tracer $\Delta^{14}CO_2$ in atmospheric observing networks for the foreseeable future. As we discuss in the paper, CO is not a perfect ffCO₂ proxy for many reasons. Therefore, we consider it the primary goal of our study to emphasise the shortcomings of the CO-proxy approach clearly. This includes showing that the use of the CO proxy requires calibration by $^{14}C$ measurements to achieve the necessary precision. Despite all the difficulties and deficits of the CO-proxy-based ffCO₂ estimation, we demonstrate in this paper and in the companion paper by Maier et al. (2023) the great potential of this method. We understand that due to our comprehensive and in-depth analysis of the problems of the CO-proxy, the impression of the measurement report may arise. However, it is precisely this detailed and dedicated investigation of the CO-proxy approach that sets this publication apart from its predecessors and makes future users of the CO-proxy approach, whether for in-situ or remote sensing approaches, aware of its potential and shortcomings. We revised the introduction of our manuscript and tried to make the focus of our manuscript clearer by addressing the points mentioned above (p. 3, l. 85-92).

1). The authors point out the disadvantage of using C-14 measurement to assess ΔffCO2, high cost, low coverage, and etc., and thus there is a need of independent tracers that are easy monitoring. The proposed tracer is CO. But in order to use CO to estimate ΔffCO2 in a site or regions, one has to first measure enough C-14 data to build a robust CO/ΔffCO2 ratio? This is sort of a loop, if there is already C-14 measurements, isn't that can be directly used to calculate ΔffCO2? Although CO measurements can be continuous, so as the derived ΔffCO2 with a known CO/ΔffCO2 ratio, would continuous ΔffCO2 record be offering

significantly more valuable information than discrete (e.g., daily or hourly average) ΔffCO2 measurements?

As mentioned above, using $^{14}CO_2$ measurements is the most direct way to estimate $\Delta ffCO_2$ concentrations. However, the sparse $^{14}CO_2$ observations lead to a small amount of $\Delta ffCO_2$ estimates, which could be used in inverse models to infer $ffCO_2$ emissions. The question is then: is the amount of $\Delta ffCO_2$ data enough for getting robust and data-driven inversion results? This is exactly what we investigated in the companion paper (Maier et al., 2023), which followed up this study. It turned out, that we indeed get no robust $ffCO_2$ emission estimates for the surroundings of Heidelberg if we use the $^{14}C$-based $\Delta ffCO_2$ estimates from the Heidelberg flasks. This can mainly be explained by the very heterogenic distribution of the $ffCO_2$ sources around Heidelberg (including several large point sources) and the shortcomings of the transport model in accurately simulating $ffCO_2$ concentrations for individual (afternoon) hours. In contrast, the continuous Heidelberg CO-based $\Delta ffCO_2$ estimates, which we derived in this study, yield robust and data-driven inversion results that could be used to investigate the effect of the Corona restrictions and to validate the seasonal cycle of the TNO $ffCO_2$ emission inventory in the main footprint of Heidelberg. Thus, the continuous CO-based $\Delta ffCO_2$ record indeed yields more valuable information than the discrete $^{14}C$-based $\Delta ffCO_2$ estimates from flasks for this urban region around Heidelberg and at the present time when the traffic and heating sectors show here similar $CO/ffCO_2$ emission ratios. Of course, if the $^{14}C$-based $\Delta ffCO_2$ had a similar temporal resolution like the CO-based $\Delta ffCO_2$, the information content of the $^{14}C$-based $\Delta ffCO_2$ would be higher than that of the CO-based $\Delta ffCO_2$. However, as this is not (yet) the case, we conclude that the usage of CO-based $\Delta ffCO_2$ to infer $ffCO_2$ emissions could be a valuable approach for other (urban) sites.
For such a CO-based $\Delta ffCO_2$ inversion, it is essential to have a reliable quantification of the uncertainties of the CO-based $\Delta ffCO_2$ estimates and an investigation of potential seasonal or diurnal cycles in the $\Delta CO/\Delta ffCO_2$ ratios. Both aspects are treated in detail in our present study, which is why it forms the basis for CO-based $\Delta ffCO_2$ inversions.
We added these main results of the companion paper in the manuscript to further motivate the CO proxy approach (p. 3, l. 94-96; p. 3-4, l. 101-103; p. 16, l. 415-421; p. 20-21, l. 571-573).

2) I am not sure if this is the authors' idea: once a robust $CO/\Delta ffCO2$ ratio was available, then perhaps this ratio is able to apply to other locations or the same location but at a different time? This is my impression of reading the manuscript, and this is perhaps the most (if not only) significance of establishing the ratio. However, as the measurement reports from the two sites indicates, the $CO/\Delta ffCO2$ ratio are different among these sites, and there are also large temporal variations. This causes doubts on the ability of the ratio to be applied to period or regions without C-14 but only CO observations. In fact, due to spatial and temporal variations in the use of fossil fuel energy, as well as the spatial and temporal variations in other sources/factors that could influence the production and atmospheric removal of CO, there is a doubt that whether the $CO/\Delta ffCO2$ ratio can stay relatively constant with time and space.

We agree, the $\Delta CO/\Delta ffCO_2$ can show large spatial and temporal differences due to the spatiotemporal variability of the $CO/ffCO_2$ emission ratios and/or non-fossil CO

contributions. This is also what we found in our study: the average $\Delta CO/\Delta ffCO_2$ ratio in Heidelberg is lower compared to that at OPE. That's why we do not recommend to apply a single $\Delta CO/\Delta ffCO_2$ ratio for different sites or times, if it is not validated by [14]C measurements at the respective sites. In our study, we discuss how many [14]C measurements should be performed to get robust $\Delta CO/\Delta ffCO_2$ ratios, which can then be used to construct a CO-based $\Delta ffCO_2$ record with high temporal resolution.

Moreover, our comparison of the observed ratios with inventory-based ratios from TNO clearly reveals that the $\Delta CO/\Delta ffCO_2$ ratios should be calibrated with [14]C measurements to avoid large biases in the CO-based $\Delta ffCO_2$ estimates. Ongoing [14]C measurements are therefore a prerequisite for monitoring changes in future $\Delta CO/\Delta ffCO_2$ ratios and continuing to apply the CO proxy approach. This information is also relevant for studies combining satellite-based CO observations and inventory-based $CO/ffCO_2$ emission ratios to derive $ffCO_2$ emissions. We made these points clearer in the manuscript (p. 4, l. 121-124; p. 20, l. 552-554; p. 21, l. 594-599).

The first concern emphasizes that no matter what one needs to measure C-14, and the second indicates the established ratio is probably not able to fulfil the intended tasks. As such, the scientific significance of these measurements is somewhat weak, and this is why I suggested that this manuscript is more like a measurement report.

Unfortunately, it is not straightforward to calculate $\Delta ffCO_2$ concentrations, which then could be used to derive top-down $ffCO_2$ emission estimates. If there were continuous [14]C measurements with high spatiotemporal coverage and low uncertainty, we would hardly need to think about using CO as a tracer for $ffCO_2$. However, this is at least not yet the case. Thus, the CO proxy approach is a way to bridge the gap between more reliable but sparse [14]C-based $\Delta ffCO_2$ estimates with temporal information derived from CO. We added this point in the manuscript (p. 3, l. 87-88).

With this in mind, we are convinced that our present study is important and has a scientific significance. We want to summarize here the main points, why we do not think that this study is just a measurement report:

1. In addition to quantifying the uncertainties of the CO-based $\Delta ffCO_2$, we conducted a synthetic data experiment to characterize and interpret the cause of these uncertainties. From this study, we were able to draw conclusions about the impact of the spatiotemporal variability of the $CO/ffCO_2$ emission ratios on the CO-based $\Delta ffCO_2$ uncertainties at an urban and a rural site.

2. We used the STILT model to also simulate $\Delta CO/\Delta ffCO_2$ ratios for both sites. We carefully compared the simulated ratios with our measurements and demonstrate that there can be substantial biases. Our study suggests that the TNO inventory may underestimate the $CO/ffCO_2$ emission ratios in the Rhine Valley. A more detailed investigation (in the revised version of the manuscript, p. 18, l. 494-504) reveals that especially the $CO/ffCO_2$ emissions from the TNO heating sector might be too low in the main footprint of Heidelberg. Such studies are important to improve the emission inventories.

3. By performing a bootstrapping approach, we provide a rough estimate on how many [14]C flasks should be collected at other sites to obtain robust $\Delta CO/\Delta ffCO_2$ ratios, which can then be used to derive CO-based $\Delta ffCO_2$ concentrations at those sites. This is an essential information if CO-based $\Delta ffCO_2$ data is to be used in inverse models to

estimate ffCO$_2$ emissions. In our follow-up study, we show that the CO-based ΔffCO$_2$ data from Heidelberg lead to robust ffCO$_2$ emission estimates, which could be used to validate the seasonal cycle of the TNO emission inventory.

From that we argue that our study has enough scientific significance to be published as a research article.

We incorporated these points in the introcution of our manuscript to provide a better overview of our study (p. 3, l. 98-106; p. 4, l. 121-124).

In addition, I have some technical suggestions that the authors can consider to use or not:

1) in the method part, please state more clearly how other sources especially the biomass burning source of CO is treated to get ΔCO; This includes the modeling of inventory based ratio.

The TNO inventory includes CO emissions from agricultural waste burning, i.e. the burning of crop residues based on a 10 year climatology. However, CO emissions from forest fires are not included. These emissions are erratic and can be obtained from e.g. the Copernicus Global Fire assimilation system (GFAS). For the central European region, forest fires are not a dominant source of CO. We clarified this in the manuscript. (p. 8, l. 220-223).

2) In 2.2.1., the defined Rflask is in fact not used but instead ΔCO/ΔffCO2 in follow-up results and discussion. I suggest to keep constant throughout the manuscript.

Thank you for this hint. We replaced R$_{flask}$ by ΔCO/ΔffCO$_2$ in our manuscript (p. 5, l. 154-155; p. 6, l. 182-183; p. 7, l. 201; p. 9, l. 247).

3) Figure 3b, could not understand how real flask measurements can be plotted in the figure, the X-axis is measurements, while the Y-axis is synthetic data which is not scaled to real measurements.

The real and synthetic data share the same X-axis and Y-axis. We labelled the X-axis with "$^{14}$C-based ΔffCO$_2$" because it refers to the $^{14}$C-based ΔffCO$_2$ observations (in case of the real observations, i.e. the black and grey dots) and to the error-prone synthetic ΔffCO$_2$ data (in case of the synthetic data, i.e. the red and orange dots), which should mimic the variability of the $^{14}$C-based ΔffCO$_2$ observations. Similarly, the Y-axis with label "ΔCO-based ΔffCO$_2$" refers to the ΔCO-based ΔffCO$_2$ observations as well as to the synthetic ΔCO-based ΔffCO$_2$ data, which were constructed by dividing the synthetic ΔCO data by a constant ΔCO/ΔffCO$_2$ ratio of 8.44 ppb/ppm. We tried to make this clearer in the caption of Fig. 3 (p. 10; l. 267-270; p. 10, l. 273-274; p. 11, l. 292-293).

References:

Maier, F. M., Rödenbeck, C., Levin, I., Gerbig, C., Gachkivskyi, M., and Hammer, S.: Potential of $^{14}$C-based versus ΔCO-based ΔffCO$_2$ observations to estimate urban fossil fuel CO$_2$ (ffCO$_2$) emissions, EGUsphere [preprint], https://doi.org/10.5194/egusphere-2023-1239, 2023.

**Replies to review 2**

We want to thank John Miller for the review of our manuscript and his helpful suggestions for improving this study. Our replies are marked in blue.

Review of Maier et al, "Uncertainty of continuous ΔCO-based ΔffCO2 estimates derived from $^{14}$C flask and bottom-up ΔCO/ΔffCO2 ratios" by John Miller

**General comments**

Overall, this is a very good study focused on developing high frequency proxies (here CO) for the estimation for the recently added fossil content of atmospheric $CO_2$ measurements. The writing is generally very good and the figures are excellent, reflecting the analysis itself. The pdf has numerous comments but I will highlight two here:

1. Although this is clear in the title, use of "Delta(CO)-based DffCO2 estimates", when used without explanation, has the potential to be highly misleading because these estimates (the atmospheric data-based ones the paper shows to be trustworthy) are still based on D$^{14}$C I'm not exactly sure of the solution, but perhaps you can employ nomenclature/notation that identifies such values as 'calibrated' by $^{14}$C.

   We agree, that this can lead to confusion. We tried to make this clearer in our manuscript, e.g. by using "proxy-based ΔffCO$_2$" in the abstract, as you suggested. However, in the main text we would like to use "ΔCO-based ΔffCO$_2$" because this is also what we used in the companion paper (Maier et al., 2023). We also added some text in the introduction, to make it clear, that the "ΔCO-based ΔffCO$_2$" is calibrated with $^{14}$C. (p. 1, l. 21-24; p. 1, l. 25-29; e.g., p. 3, l. 70-72; p. 3, l. 96-98; p. 5, l. 152; p. 14, l. 375; p. 15, l. 408-409; p. 21, l. 595-597)

2. I have a few questions about the TNO inventory that could benefit by a bit more investigation and explanation. First, it appears that TNO includes biofuels such as wood. But what about ethanol and biodiesel? Generally, can the fossil components of the TNO inventory be isolated for a more direct comparison with $^{14}$C-based observations? Second, in investigation of the point source impacts for Heidelberg, can you transport the non-point-source sectors to see how much the match to data is improved – i.e., is the mismatch mainly due to the ratio of (dilute) point to area sources in TNO or mainly due to incorrect emission ratios for the area sources?

   Thank you for your suggestions. The TNO inventory distinguishes between fossil fuel and biofuel CO$_2$ and CO emissions. Thus, TNO includes emissions from wood-fired heating as well as biofuel emissions from the traffic sector. For the traffic sector this is based on national reporting of shares of ethanol in gasoline and/or biodiesel in diesel. Moreover, there is also data on the amount of biomass co-fired in coal-fired powerplants. This is also in the inventory as CO2 from biofuel.

We carried out both of the simulations you suggested (see Fig. 1 below). The magenta dots in Figure 1a show the contributions from the non-point-source sector only. They lead to an average ratio of about 6 ppb/ppm, which is well below the observed ratio (grey crosses). From that we conclude that TNO might underestimate the ratio of the area source emissions in the Rhine Valley. Furthermore, a potential wrong dilution ratio between point sources and area sources (both with correct emission ratios) can't explain the differences between observed and modelled $\Delta CO/\Delta ffCO_2$ ratios, because both, the average modelled ratio from only-area sources (6.0 ppb/ppm) and the ratio from only-point sources (1.2 ppb/ppm) are below the observed ratio of 8.44 ppb/ppm.

Fig. 1b shows the simulated contributions from the traffic (orange) and heating (cyan) sectors. The traffic (biofuel plus fossil fuel) sector leads to an average $\Delta CO/\Delta ffCO_2$ ratio of 7.72±0.08 ppb/ppm ($R^2$=0.93), which is less than 10% lower compared to the observed average ratio of 8.44 ppb/ppm. However, the heating (wood plus fossil) sector leads to a much lower average $\Delta CO/\Delta ffCO_2$ ratio of 3.36±0.09 ppb/ppm ($R^2$=0.65). This might indicate that the $CO/ffCO_2$ emission ratios from the TNO heating sector are too low in the main footprint of Heidelberg. This could be explained, for example, by an incorrect distribution of the use of fossil and bio fuels in the heating sector.

We included these plots and a discussion in our manuscript. (Fig. 4b and A3 in the manuscript; p. 13, l. 328-329; p. 18-19, l. 492-516)

[Figure]

Figure 1: $\Delta CO$ and $^{14}C$-based $\Delta ffCO_2$ observations from the Heidelberg flasks (in grey) and (a) simulated non-point-source $\Delta CO$ and $\Delta ffCO_2$ contributions for the flask events (magenta), and (b) simulated $\Delta CO$ and $\Delta ffCO_2$ contributions from the traffic (orange) and heating (cyan) sectors for the flask events.

**Specific comments**

Suggested edits and comments are embedded in the manuscript .pdf. Blue highlights indicate those that are language oriented and yellow for science/conceptual issues.

Thank you for your comments. We directly replied to them in the .pdf document. We also list them here together with our responses and the changes we have made to the manuscript:

| Reviewer comment | Our response |
|---|---|
| you need to say what Delta is. | Done. (p. 1, l. 20-21) |
| say briefly what background is | Done. (p. 1, l. 24-25) |
| this is maybe too optimistic or at least too broad, because there are aspects of the flask-based Cff that contain bias (such as background). I suspect that you are saying that relative to "direct" Cff, the CO-proxy Cff are unbiased. You might want just delete 'bias-free' at this point and a few sentences later refer to the calculated bias (see note below). | Thank you for this suggestion. We deleted 'bias-free' and added a sentence with the calculated (small) bias. (p. 1, l. 26-28; p. 16, l. 414-415) |
| I think this nomenclature is misleading because both of these rely critically on D14C measurements; the former is not purely CO-based. I would consider something along the lines "direct" CO2ff and "proxy" CO2ff | We agree, and adopted your suggestion here. (p. 1, l. 25-29) |
| RMSD is potentially useful but because it combines the standard deviation and the mean bias of the residuals it merges 'bias' and 'uncertainty'. I think it's important to report both the mean bias and variance/s.d. of the differences because these are telling you two different things. | Done. (p. 1, l. 26-29) |
| obviously there are lots of choices here but I will point out the Miller et al. 2020 PNAS, presents a dataset of over 400 CO2ff values for Los Angeles. | Thank you for this reference. We have included it. (p. 2, l. 43) |
| and | Done. (p. 2, l. 48) |
| *typically* co-emitted. Deisel engines and powerplants emit very small amounts of CO. | Done. (p. 2, l. 52) |
| in Turnbull et al. 2006 we defined the term R_CO. This had a slightly different definition originally and was then revised to your definition in Turnbull et al, 2011. R_CO might be a useful shorthand notation here. | Thanks for this suggestion. However, we decided to stick with our notation to be consistent with the title of this study (and the usage of R_CO in the title might be less intuitive). |
| "allows us to" or "allows one to" | Done. (p. 3, l. 70) |
| please also cite Turnbull et al, 2011 (Sacramento aircraft study) as an example of applying high frequency 14C-calibrated CO as a proxy. | Thank you for this nice reference. We added a few sentences about this study in the manuscript. (p. 3, l. 72-76) |
| reduced | Done. (p. 3, l. 82) |

| | |
|---|---|
| in the calculation of Delta CO, the oxidation of CH4 will have almost no impact because this term's impact on the background and downwind sites will be almost identical. also CH4 is a VOC, so you should either write "VOCs" or "CH4 and non-methane VOCs" | Done. (p. 4, l. 118) |
| 500 m from | Done. (p. 4, l. 132) |
| I know it is very common for all of us to say "flask samples" but the sample is really an "air sample" captured by a flask. Obviously that's a lot to write every time. However, here's an opportunity to write something more accurately and efficiently.
        "hourly air samples are collected ...with and automated flask sampler". Not at all a critical issue... | Done. (p. 4, l. 139-140) |
| It would be very useful to provide an estimate (even with large variability/uncertainty) of the nighttime PBL inversion height, as well as the fact that Heidelberg is in quite a steep valley so that estimating this quantity, proportional to the volume into which fluxes mix, could be quite difficult to estimate.  The discussion of these impacts could be presented when interpreting results. | Thank you for this comment. We included a paragraph about this issue in the discussion section. However, we assume that potential errors in the STILT mixing heights might affect both, the simulated CO and ffCO2 concentrations similarly. That's why we would expect only a minor impact of inaccurate STILT mixing heights on the modelled CO/ffCO2 ratios. (p. 20, l. 556-562) |
| mole fractions | Done. (p. 5, l. 145) |
| This is mentioned above already, but if you are going to take the time to explain this in the next part of the sentence, mention that it is measurement of d13C that allows for correction of fractionation; also "in per mil" can be dropped. | Done. (p. 5, l. 148-150) |
| if you are using "Delta", I think you can drop excess, as excess is what the Delta signifies. | Done. (p. 5, l. 156) |
| The s.d. of residuals to the fit is...  (the fit itself produces a covariance matrix describing the uncertainty of the fit, but I don't think that is what you are describing) | We refer to the fit uncertainty. We clarified it in the text. (p. 5, l. 163-164) |
| So you used Kresin as an alternative background?  I would say that more explicitly to improved clarity. | No, we always used MHD as our background. In Maier et al. (2023b), we conducted a model study to estimate the representativeness bias and uncertainty of the MHD background if this marine background is used as a background for |

| | observations at the eastern site Kresin. As Kresin is located further east of Heidelberg, it might be stronger influenced by continental easterly air masses than Heidelberg. That's why we use this estimate as an upper limit for the MHD background representativeness uncertainty in our study. We tried to make this clearer now in the manuscript. (p. 6, l. 166-172) |
|---|---|
| I don't think it's appropriate to use 'unbiased' here, even though the appendix shows no bias from the fitting technique. In the real world there are other biases you are not capturing that weren't modeled in your synthetic tests (e.g. seasonal and spatial correlations in R_flask_i and magnitude of CO2ff (that will influence the slope). Describe the narrow sense of bias you're referring to here, or otherwise, just leave out 'unbiased' as this is described later. | Done. (p. 7, l. 201) |
| if you truly believe the uncertainty estimate of the regression method of 0.17 ppb/ppm for R_flask, which is the quantity you're interested in, why should you worry about the R^2 value? | We agree, that the R^2 should decrease for increasing slope uncertainties. However, we would like to show here also the R^2 values to more intuitively illustrate that the correlation between CO and ffCO2 is much smaller in summer compared to winter. |
| are you referring to the R^2 value as the basis for 'difficult'? | Yes, 'difficult' because of poorer correlation. We added this information in the text. (p. 11, l. 275-277) |
| can you also plot the impact of all the non-point sources from TNO? This would help you to tell the difference between 1) TNO having its area source emission ratios incorrect, and 2) having the ratio between point sources and area source (both with correct emission ratios) incorrect. | Thanks a lot for this suggestion. We included this plot. The non-point sources from TNO lead to an average ratio of about 6 ppb/ppm, which is well below the observed ratio. From that we conclude that TNO might underestimate the ratio of the area source emissions in the Rhine Valley. And a potential wrong ratio between point sources and area sources (both with correct emission ratios) can't explain the differences between observed and modelled ratios, because both, the average modelled ratio from only-area sources (6.0ppb/ppm) and the ratio from only-point sources (1.2ppb/ppm) are below the observed ratio. (see Fig. 4b in the revised |

| | manuscript, and p. 13, l. 308-309; p. 12-13, l. 321-330; p. 18, l. 492-494) |
|---|---|
| with CO2ff up to 10 ppm relative to MHD I don't think you can call this a 'remote site'. Maybe 'rural' or "more remote'? | You are right. We have changed this. (p. 13, l. 342; p. 21, l. 582+588) |
| why would this impact the CO2ff calculation.  if you're using total CO2, of course, but not with 14CO2.  Instead, given that there is not a large fossil co2 emissions seasonal cycle, I suspect that a big part of this phenomenon is the much higher dilution volumes (i.e. PBL height) in summer than winter. | We meant with "biospheric fluxes" the contribution from non-fossil CO sources (e.g. oxidation of VOCs) during summer, which impacts the CO concentrations but not the ffCO2 concentrations, and thus could lead to worse CO/ffCO2 correlation. We deleted "biospheric fluxes" here since it is redundant, as we also mention "non-fossil CO sources". We agree, that a big part of the worse correlation in summer can be explained by the smaller signals in summer due to the higher mixing volumes. This is also what we found in the synthetic study shown in Appendix A1. (p. 15, l. 401-404) |
| This is a bit confusing to me.  I don't think any realistic sampling plan would be random; rather the frequency would be varied from 1- 14 days (e.g.), but the period would be steady.  That would be a much more interesting result.

you could still introduce a random element by changing the days of the week for each iteration. | One aim of our study was to investigate if there is a potential seasonal or diurnal cycle in the CO/ffCO2 ratios at Heidelberg. To do this, we needed a flask pool, which is as representative as possible. Therefore, we collected flasks during very different conditions, e.g. in the rush-hours, in the afternoon, during night, or consecutively within a synoptic event. Because of this very irregular flask sampling strategy, we cannot conduct the experiment you suggested.
However, if one already knows that there is no diurnal cycle in the CO/ffCO2 ratios or if one only wants to have proxy-based ffCO2 concentrations during the afternoon, it might be sufficient to collect afternoon-only flasks in a regular manner. Then your experiment could be conducted and might indeed reveal interesting information. |
| the test I suggested above could help evaluate this, although transporting the non-point-sources would also including building emissions. | We have rewritten this paragraph slightly and added a discussion of your proposed test here. (p. 18-19, l. 492-516) |
| Does TNO not allow the separation of biofuels like wood, and ethanol and biodiesel for the road sector? | Yes, TNO allows the separation between fossil fuel and biofuel for each of the emission sectors. This is a good idea. We simulated the CO/ffCO2 ratios from the |

| | TNO traffic and heating sector separately and added a plot (Fig. A3) and its discussion in the manuscript. (p. 18-19, l. 492-516) |
|---|---|
| and a negligible | Done. (p. 19, l. 532) |
| can be most easily explained | Done. (p. 19, l. 532) |
| wouldn't that create a positive intercept? | Yes, your are right. We deleted this. (p. 19, l. 533) |

References:

Maier, F. M., Rödenbeck, C., Levin, I., Gerbig, C., Gachkivskyi, M., and Hammer, S.: Potential of $^{14}$C-based versus $\Delta$CO-based $\Delta$ffCO$_2$ observations to estimate urban fossil fuel CO$_2$ (ffCO$_2$) emissions, EGUsphere [preprint], https://doi.org/10.5194/egusphere-2023-1239, 2023.